# Quantitative evaluation of therapeutic effect of silver needle thermal conduction therapy on myofascial trigger point

Yue Qin[1,2] , Yue Wang[1,3] , Chunxin Wo[1‡], Zhenmin Wang[4‡], Zilong Yu[1,2‡], Yuanxin Huang[2]*, Lin Wang[2]*

1 Clinical Medicine College, Guizhou Medical University, Guiyang, Guizhou, China, 2 Department of Algology, The Affiliated Hospital of Guizhou Medical University, Guiyang, Guizhou, China, 3 Department of Algology, Beijing Jishuitan Hospital Guizhou Hospital, Guiyang, Guizhou, China, 4 Department of Imaging, The Affiliated Hospital of Guizhou Medical University, Guiyang, Guizhou, China

☯ These authors contributed equally to this work.
‡ CW, ZW and ZY also contributed equally to this work.
* wanglin0949@yeah.net (LW); 18985116689@sina.com (YH)

## Abstract

### Background

In clinical practice, silver needle thermal conduction therapy has a definite effect on myofascial pain syndrome (MPS). However, there is a lack of objective evidence to evaluate the efficacy of this therapy. This study aimed to assess the effectiveness of silver needle thermal conduction therapy on MPS rats by objective therapeutic index.

### Methods

MPS model was established by blunt strick combined with centrifugal running training. MPS rats were divided into model and treatment groups, with a synchronized control group. The model group received no treatment, whereas the treatment group underwent silver needle thermal conduction therapy. The T2 value and stiffness values were evaluated by magnetic resonance imaging and sound touch elastography. The ultrastructure of muscle mitochondria was examined using transmission electron microscopy, and the Silent mating type information regulation 2 homolog 3 (SIRT3) expression level was evaluated by western blotting.

### Results

T2 values and elastic modulus values in the treatment group were lower than those in the model group, and there was no difference between the treatment group and the control group. Mitochondrial damage was observed in the model group, and the degree of mitochondrial damage in the treatment group was less than that in the

**Editor:** Jiajia Ye, Rehabilitative Hospital Affiliated to Fujian University of Traditional Chinese Medicine: Fujian University of Traditional Chinese Medicine Affiliated Rehabilitative Hospital, CHINA

**Data availability statement:** All files are available from the DRYAD database, URL: http://datadryad.org/stash/share/yrS4Eqp84CYX-kLfTqmg-D3tQuxMxCnnocczZoxkvpSI. (DOI: 10.5061/dryad.7m0cfxq5z).

**Funding:** This work was supported by the National Natural Science Foundation of China (Grants No: 82160226/H2902) , the Natural Science Foundation of China (Grants No: 82060811/H2902), Guizhou Province Science and Technology Plan Project (Grants No:Qianke He Foundation -ZK[2021] General 508), Guizhou Province Science and Technology Plan Project (Grants No:Qianke He Foundation -ZK[2023]General 370), Guizhou Administration of Traditional Chinese Medcine (Grants No: QZYY-2021-123) and the Guizhou Province Science and Technology Plan Project (Grant No: Qianke He Foundation -ZK[2024] Key 036). Yuanxin Huang, Yue Qin, and Lin Wang are the recipients of the funding awards. The funders had no role in study design, data collection and analysis, decision to publish, or preparation of the manuscript.

**Competing interests:** The authors have declared that no competing interests exist.

model group. SIRT3 expression in the treatment group was down-regulated compared with the normal group, but up-regulated compared with the model group.

## Conclusion

The silver needle thermal conduction therapy demonstrates the ability to reduce muscle inflammation and stiffness and facilitate the repair of damaged muscle mitochondria.

## Introduction

Myofascial Pain Syndrome (MPS) is a prevalent regional pain disorder that affects individuals of all age groups. It is characterized by the presence of trigger point (TrP) within the muscles or fascia [1]. The Global Burden of Diseases, Injuries, and Risk Factors Study 2017 confirmed the significant contribution of osteoarthritis, low back pain, neck pain, and other musculoskeletal disorders to the global years lived with disability, comprising 40–50% of the economic burden for work-related disorders in the European Community [2]. While not life-threatening, these disorders significantly impact quality of life and psychosocial well-being.

Silver needle thermal conduction therapy has evolved from traditional acupuncture. It combines the acupuncture effect with the thermal effect to play a therapeutic role. The silver needle thermal conduction therapy and the dry needle therapy differ in their treatment methods.: 1. Different treatment sites: Silver needles are based on the anatomical structure of the painful area and are used for treatment at the starting point, ending point or course of the muscle, while dry needles are used for treatment at the trigger point. 2. Different needling depths: silver needles reaches the periosteum. 3. Different thicknesses: The silver needle has a thicker body, with a diameter of approximately 1.1 mm and a length ranging from 110 to 150 mm. The dry needle has a diameter of about 0.3 to 0.5 mm and a length of about 40–50 mm [3,4]. The length and diameter of the silver needles are approximately three times that of the dry needle. 4. Different number of treatments: silver needle thermal conduction therapy only requires a single treatment, while dry needle therapy needs one or multiple sessions [5].

Due to the lack of more objective evidence and basic research on this therapy, it is not known by most scholars. Meanwhile, Clinical assessment of TrPs is subjective and exhibits poor inter-rater reliability [6]. The absence of a widely accepted objective reference standard hampers the precision of clinical diagnosis, therapeutic management, and the generation of high-quality scientific evidence [7].

Sound touch elastography (STE), the latest shear wave elastography of ultrasonography, is an imaging technique that displays 2D elastic images of region of interest (ROI) in real time and provides elastic values. It can display the absolute value of tissue stiffness noninvasively and quantitatively and has high accuracy [8]. STE imaging technology compensates for the poor repeatability caused by manually selecting ROI to measure the elasticity value of lesions in the past, and also avoids

errors caused by human operation. STE is now widely used to measure the hardness of liver, thyroid, breast and other tissues.

Magnetic resonance imaging (MRI) is recognized as the reference standard to assess muscle morphology due to its ability to visualize soft tissues with excellent contrast, high resolution, and multiplanar assessment [9]. T2 mapping is used to quantify T2 values. The pathological processes of skeletal muscle (inflammation, tissue damage, etc.) can all lead to changes in T2 values. The T2 value of skeletal muscle can serve as a sensitive indicator reflecting changes in the internal structure of skeletal muscle [10–11].

The causative mechanisms of MPS remain under debate. Most people currently consider TrPs to be the main characteristic of members of Congress. The widely accepted mechanism for the formation of TrPs is the "TrPs synthesis hypothesis" proposed by Simons. This hypothesis suggests that under various nociceptive stimuli, abnormal motor endplate function may occur after muscle fiber injury, resulting in excessive release of acetylcholine in the motor endplate area, continuous depolarization of the postsynaptic membrane of neuromuscular junction, lack of ATP reabsorption of $Ca^{2+}$ in the sarcoplasmic reticulum, and further release of acetylcholine in excess $Ca^{2+}$ resulting in sustained myotome contraction. This contraction is not controlled by nerves, and local muscle contracture nodules are formed. The continuous contraction of muscle fibers causes an energy crisis, aggravates the contraction of muscle fibers, and ultimately forms MTrPs. MTrPs compress local blood vessels, reducing the supply of nutrients and oxygen, releasing pro-inflammatory mediators and aggregating inflammatory cells leading to pain. With emerging evidence implicating mitochondrial dysfunction in the regulation of MPS pathways. Silent mating type information regulation 2 homolog 3 (SIRT3), an integral member of the Sirtuins family localized to mitochondria, can regulate the energy metabolism and function of mitochondria [12].Mitochondrial morphology and function are closely related. Electron microscopy is a golden index for observing and analyzing mitochondrial structure. Transmission electron microscopy (TEM) is a powerful tool for mitochondrial morphological examination. The combination of SIRT3 and transmission electron microscopy can be used to evaluate mitochondrial function.

Given this background, our study aimed to quantitatively evaluate the influence of silver needle thermal conduction therapy on MPS by T2 mapping, STE and other objective indicators. This study is anticipated to provide a more tangible basis for the application of silver needle thermal conduction therapy in MPS treatment.

## Materials and methods

### Animals and ethical statement

15 healthy wild type adult Sprague-Dawley male rats (6~8 week old) were used in our investigation. Rats were individually housed in galvanized wire mesh cages with free access to food and water and kept on an alternate 12-hour light/dark cycle. The laboratory room temperature was kept between 21–25°C and the humidity was maintained at 50% to 55%. According to the guidelines provided by the International Society of Animal pain Studies, each rat was adequately accommodated and cared for to minimize pain and discomfort as well as the number of animals used. Animal experiments have been approved by the Animal Care Welfare Committee of Guizhou Medical University (NO.2201171)

### Experimental design

The total sample size was calculated according to the resource equation calculation method. The minimum sample size was 15 and the maximum sample size was 24. A single rat as an experimental unit. 15 rats were randomly allocated into three groups according to the random number method: control group (n = 5), model group (n = 5), and treatment group (n = 5). The rats in control group were maintained under identical conditions to the other groups, without any interventions, and were sacrificed along with the other groups. Chronic MPS models were induced in the model group through impact combined with exercise fatigue. Rats with fractures, hematoma, or no change in pain threshold were excluded. Rats with reduced pain threshold were included in subsequent experiments. After successful model induction, rats in treatment

group underwent silver needle thermal conduction treatment. The rats were sacrificed 14 days post-completion of the silver needle thermal conduction treatment. In each group, electromyographic activity of the right medial femoral muscle was recorded.

The changes of TrPs region were observed by SEA and HE staining. A 3 T MRI scanner was employed for MRI examination with a T2 mapping protocol. Ultrasound elastography was performed using the STE system. Transmission electron microscopy was also employed to detect the mitochondrial morphology under the ultrastructure of the TrPs parts of the rats in each group. The rats were killed after anesthesia. Finally, western blot analysis was used to assess the expression of SIRT3. The implementation of the experiment, outcome evaluation and data analysis were completed by different person. The intervention schedule is shown in Fig 1.

## Establishment of the chronic MPS model

The model establishment process consisted of an intervention period lasting eight weeks and a subsequent recovery period lasting four weeks. Three days prior to the experiment, all rats were acclimated to a treadmill (WI32812 multi-channel running, Dongxiyi Technology Co., Ltd. Beijing) for 15 min and accustomed to locomotion on the treadmill.

Rats were subjected to blunt strikes on the first day of each week during the intervention period. Anesthesia was induced with 3 ml/kg of 1% sodium pentobarbital (Shanghai New Asiatic Pharmaceuticals Co., Ltd, Shanghai, China) into the abdominal cavity. A single blunt blow was performed on the right medial femoris muscle of rats under anesthesia. The rats were fixed at the bottom of the homemade striking device, and a 1000g wooden stick on the striker was dropped onto the right medial femoral muscle from a height of 20 cm with a kinetic energy of 2.352 J. The contact area between the stick and the muscle is 1 cm × 1 cm. On the second day, centrifugal running training was carried out, set to −16°downhill running mode at a running speed of 16 m/min. The duration was 90 min, and the rats were driven away by sound and electricity. All the rats in the model and treatment groups were subjected to training on the centrifugal treadmill. During the recovery period, no interventions were conducted. Rats resumed normal daily routines with routine feeding for four weeks.

## Silver needle thermal conduction therapy

Rats in treatment group were administered abdominal anesthesia (3 ml/kg of 1% sodium pentobarbital solution). The right medial femoral muscle was identified by palpating a taut band. This area was designated as the therapeutic target for silver needle thermal conduction therapy. The length of the silver needle is 10 cm and the insertion depth of the silver

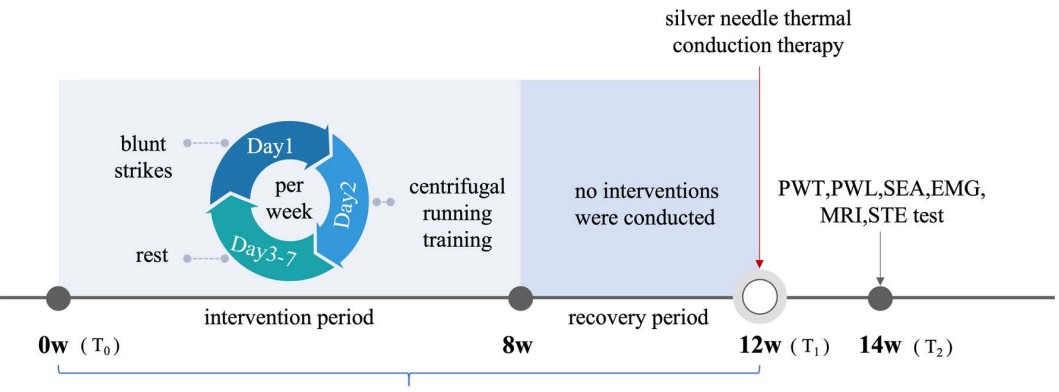

**Fig 1. Experimental design of the study and MPS model establishment.** MPS: myofascial pain syndrome; EMG: Electromyography; MRI: Magnetic Resonance Imaging; STE: sound touch elastography; SEA: spontaneous electrical activity.

needle is about 1–2 cm. Silver needle heating instrument conducts heat to the needle body by connecting with the tail end of the needle body.The heating temperature of the silver needle thermal conductor was set to 110°C. After heat dissipation through the exposed needle body, silver needle body at the skin injection point does not exceed 42°C, and the heating time was 15 min. Following needle extraction, the injection site was disinfected with 75% alcohol, covered with a sterile dressing, and normal feeding resumed.

## Pain assessment

The mechanical paw withdrawal threshold (MWT) and the thermal paw withdrawal latency (TWL) of rat models were measured by Von-Frey filaments and thermal radiometer to evaluate mechanical and thermal allodynia. The baseline MWT and TWL were measured before the MPS model establishment (T0) and at 12 (T1) and 14 (T2) weeks after model establishment. For MWT measurement, the von Frey fibers (Shanghai Yuyan Scientific Instrument Co., Ltd, China) were used to vertically stimulate the center of the rat hind paw with increasing intensity. The rat quickly flinched or licked the paw indicated a positive withdrawal reaction, and then the adjacent decreasing intensity was selected to give a stimulation. If the withdrawal reaction was negative, the adjacent fibril with increasing stimulation intensity was used to give a stimulation. Until the withdrawal reaction was positive, and there are 3 positive withdrawal reactions within the 5 consecutive stimuli, the von Frey fibers value was defined as MWT.

For TWL measurement, the hind paw of the rat was stimulated with an automatic Plantar Test (PL-200, Chengdu Taimeng Software Co., Ltd, China) and its thermal threshold was measured. The rat was placed on clear glass, covered with a transparent cover, and the infrared heat heat source was applied to the surface of the hind paw. TWL was defined as the time interval between the start of thermal stimula- tion and the paw withdrawal. Each rat was recorded 3 times with an interval of at least 5 min between the two adjacent measurements.

## Identification of TrPs with Electromyography (EMG)

Abdominal anesthesia (3 ml/kg 1% sodium pentobarbital solution) was administered to anesthetize and immobilize rats. A needle electrode was inserted into the enlarged knot, with a reference electrode near the right tibia. EMG signals were recorded in the resting state using an electromyograph (SN: ME098, Shanghai Haishen Medical Electronic Instrument Co., Ltd.).

## MRI test

After the modeling process, on the 14th day post-silver needle thermal conduction therapy, all rats were anesthetized for approximately 5 min before undergoing MRI examination using a 3T-MRI scanner (Philips MR Ingenia Elition 3.0T, Royal Dutch Philips Electronics Ltd.Netherlands). Rats were fixed in supine position with knee flexion using a 3.0T 8-channel receive rat RF coil (Wuxi Hezi Medical Technology Co., LTD, China). The image stack was centered at the proximal medial femoral level, and blankets were covered for warmth. Scanning parameters were as follows: sequence-Multi echo spin echo (SE), TR 2000ms, TE 13ms, Flip angle 90°, FOV 80mmx80mm, Slices 66 layers, Slice thickness 0.6 mm, Slice gap 0.06 mm, Acquisition time 5 min 24s. Based on the above acquisition parameters, the T2 values were derived.

## MRI imaging analysis

MRI images were post-processed using a Philips MR Ingenia Elition 3.0T MRI self-contained system. The T2 values of the right medialis muscle region were measured in the T2-mapping sequence. Three manually drawn ROI covered the right medial femoral region and the corresponding region of the contralateral hind limb, with mean T2 values calculated. Skeletal, vascular, fatty, or skin areas were avoided on ROI. All rats were measured using the same ROI. After each MRI examination, data analysis was performed by a radiologist experienced in MRI.

## STE test

The STE test was performed on rats in the anesthetized. TrPs of the right medial femoral muscle in model group were palpated and marked by a clinician with palpation expertise. An EMG examination is performed at the labeled area and the precise location where muscle twitching or spontaneous electrical activity is detected is taken as a TrP. Corresponding positions on the contralateral hind limb were also marked. Rats in control group and treatment group were marked at the same position. STE test was performed using ultrasonography (Resona R9, Mindray, Shenzhen, China) with a linear 14–3 transducer at 3–14 MHz and an elastic modulus value ranging from 0–200 kPa. In gray-scale ultrasound, the femur was used as a marker in the in-plane technique to locate the medial femoral muscle and then rotated by 90°. Meanwhile, in the out-of-plane technique, the elastic imaging mode was turned on. The imaging area was frozen after the sampling frame image was stabilized. The imaging area appeared in different colors depending on hardness. The ROIs were chosen by an experienced radiologist. A circular region with a diameter of 2 mm is selected in the red region as the TrP ROI. Adjacent area at the same muscle fiber level was selected as the adjacent ROI. The parameters and selection method of ROI referred to the previous experiments of Bubnov and Lv [12,13]. For images with a uniform distribution of elastic modulus, the ROI with the larger mean value was analyzed as the TrP area, while the other served as the adjacent area. The maximum, minimum, and mean of the elastic modulus were obtained for each ROI. Measurements were repeated three times at each location, and mean values were taken for statistical analysis.

## TEM test

Following the MRI test, rats were sacrificed with 1% sodium pentobarbital solution (3 ml/kg i.p). The right medial femoral muscles were excised, divided into three parts, and one part was cut into 0.3 mm. The samples were fixed overnight with 2.5% glutaraldehyde (Solarbio, Beijing, China) at 4°C. Following PBS rinses (three times for 10 minutes each), samples were fixed with 1% osmium tetroxide (Solarbio, Beijing, China) at room temperature for 1 hour and then embedded with 10% gelatin (Solarbio, Beijing, China). Further fixation with glutaraldehyde at 4°C for 1 hour was followed by dehydration using increasing concentrations of ethanol solution (Solarbio, Beijing, China) (30%, 50%, 70%, 90%, 95%, 100%, 100%, 100%). After immersion and embedding with epoxy resin, slices were made using a Leica UC6 ultra-thin slicer. Finally, under conditions of 110 kV, the samples were observed and photographed using a TEM (Talos F200X S/TEM, Thermo Fisher Scientific, Czech Republic).

## Hematoxylin and Eosin (HE) staining

Another part of the right medial femoral muscle was used for HE staining. Prior to immunostaining, 5-μm sections of the right medial thigh muscle tissue were dewaxed in xylene, rehydrated through decreasing concentrations of ethanol, and washed in PBS. Sections were then stained with HE staining solution (Solarbio, Beijing, China). After staining, sections were dehydrated through serially increasing concentrations of ethanol and xylene (Solarbio, Beijing, China). The morphology and arrangement of the muscle fibers were observed under an optical microscope at 100 times their actual size.

## Western blotting

A section of the rat's right medial femoral muscle tissue, obtained post-MRI examination, was stored at −80°C. The tissue was weighed, cut, and subjected to homogenization, lysis, and centrifugation. The supernatant obtained after these processes were utilized after the addition of cell lysis buffer. The protein was quantified using the BCA method, and the protein samples were electrophoresed and separated through SDS-PAGE gels, and subsequently transferred to a PVDF membrane. Following removal of the transfer solution on the PVDF membrane, blocking was carried out with 5% skim milk for 1 hour. After washing, the membrane was incubated overnight with primary antibodies against SIRT3 (1:3000, Proteintech: 10099–1-AP) and GAPDH (1:50000, Proteintech: 60004–1-Ig) at 4°C. The next day, membranes were

washed three times with Tris-buffered saline-Tween-20 (TBST) at room temperature and incubated with goat anti-mouse-HRP secondary antibody (1:1000, Proteintech: SA00001−1) for 1 hour at room temperature. After washing, electroche-miluminescence imaging was performed, and images were collected. The gray value of the protein bands was analyzed using Image J software, with the gray value of the GAPDH band used to calculate the relative expression level of SIRT3.

## Statistical analysis

All statistical analyses were conducted using GraphPad Prism 9 software. Continuous data were expressed as mean ± standard deviation (mean±SD). The normality and homogeneity of variance for continuous data were assessed using the Shapiro-Wilk test and Levene test, respectively. Group differences were analyzed using One-way ANOVA, followed by Tukey's and Games-Howell's multiple comparison tests. Within-group differences were assessed using paired samples t-test. Statistical significance was defined as $P < 0.05$.

## Results

### Pain assessment of MPS rats

There was no significant difference in average MWT and TWL among all groups at T0 (p > 0.05). Compared with control group, MWT and TWL in model group and treatment group decreased at T1 (P < 0.001). There was no significant difference between model group and treatment group (p > 0.05). At T2, PWT and PWL in treatment group were up-regulated compared with those in model group (p < 0.001) (Fig 2).

### EMG manifestations of MPS rats

Upon insertion of the EMG electrode into the enlarged knot, the model group rats exhibited a local twitch response and spontaneous electrical activity (SEA), indicating the presence of TrPs at the injury site in the right medial femoral muscle. On the other hand, no local twitch response or SEA was observed in the control group (Fig 3). 5 rats were tested in each group.

### Pathomorphological changes by HE staining

In the control group, muscle transverse sections revealed a uniform, round, or irregular structure, which was arranged closely and regularly. The transverse section of muscle tissue in the model group showed atrophy, degeneration of muscle

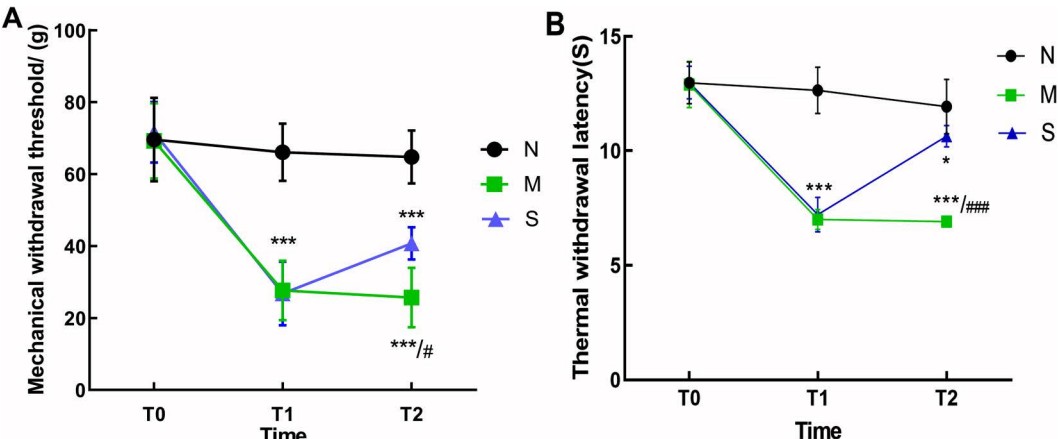

**Fig 2. Comparison of pain threshold (n = 5).** A: Comparison of mechanical withdrawal threshold in each group. B: Comparison of thermal withdrawal latency in each group. Compare with control group,****P < 0.001; Comparison between model group and control group #P < 0.05; # # # #P < 0.001.

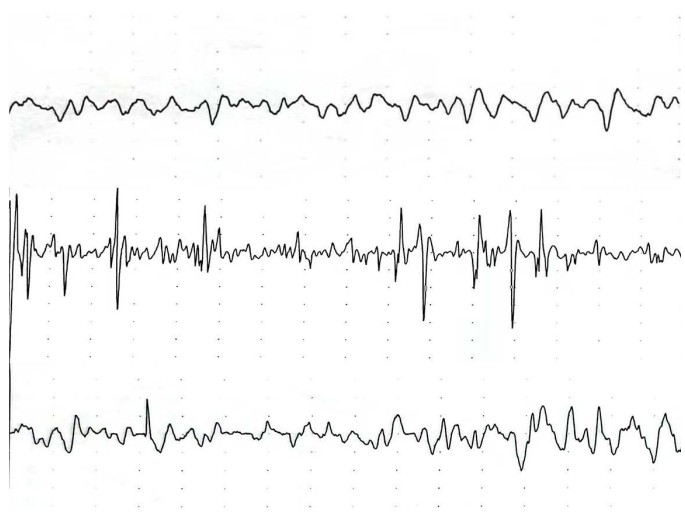

**Fig 3. EMG of the right medial femoral muscle (n = 5).** EMG: Electromyography; N:control group; M: model group; S:treatment group.

fibers, and varying sizes of elliptical and round-shaped muscle fibers. In the treatment group, we found a small number of muscle fibers that were slightly atrophied and denatured, maintaining a shape akin to that of the control group (Fig 4). 5 rats were tested in each group.

## MRI findings

The measured T2 values of the medial femoral muscles were analyzed. In control group, T2 values were uniformly distributed with a relatively low magnitude (Fig 5). The model group demonstrated significantly higher T2 values in the

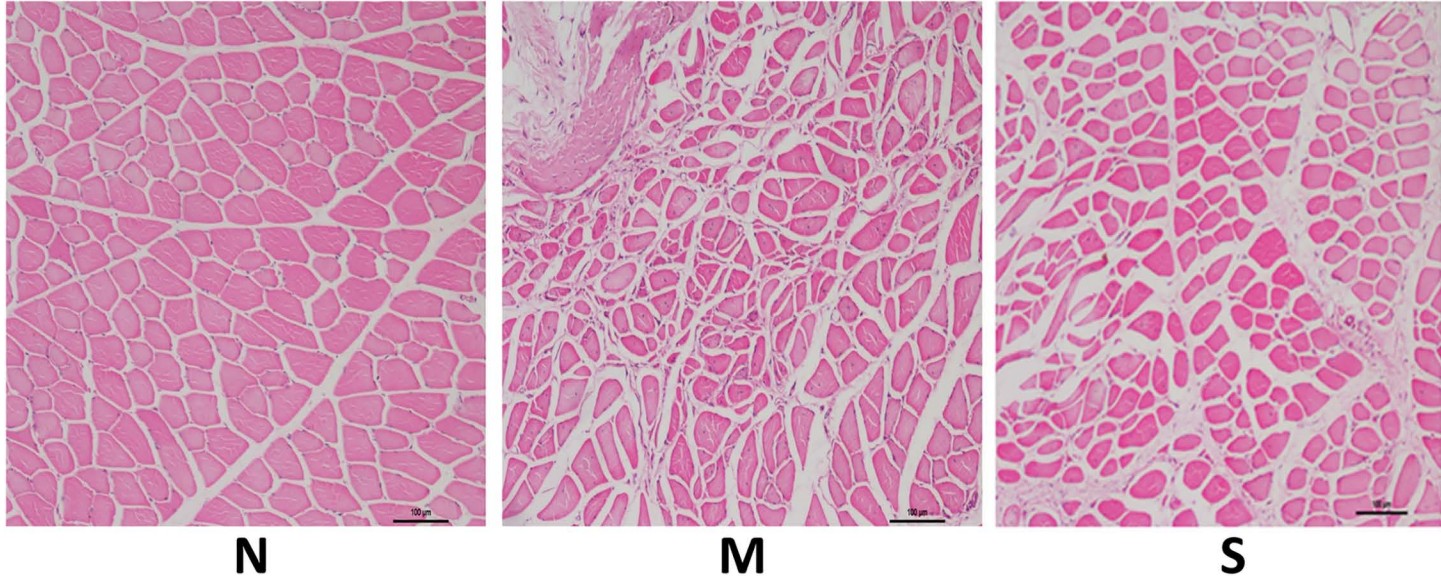

**Fig 4. HE staining of right medial femoral muscle (n = 5).** Scale bar 100μm. HE: Hematoxylin and Eosin; N:control group; M: model group; S:treatment group.

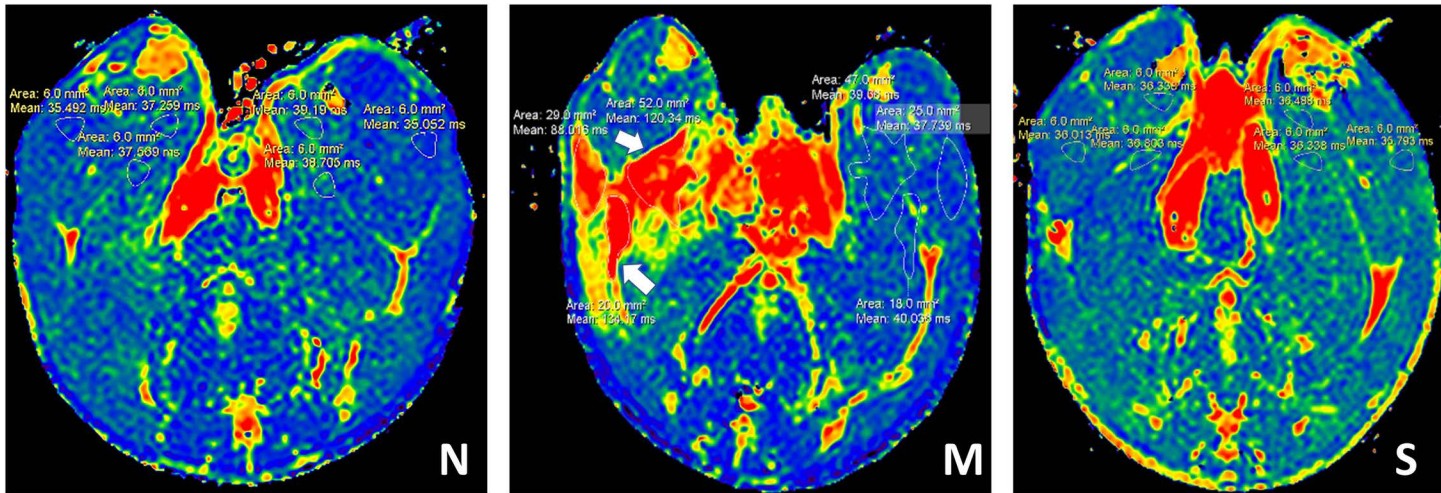

**Fig 5. MRI results for T2 mapping sequence of medial femoral muscle (n = 5).** The white arrow is the focal region of the TrP. MRI: Magnetic Resonance Imaging; TrP: trigger point; N: control group; M: model group; S: treatment group.

right medial femoral muscle compared to the control group (P < 0.001). Following silver needle thermal conduction therapy in the treatmnet group, T2 values in the right medial femoral muscle were significantly lower than in the model group (P < 0.001), with no statistically significant difference compared to the control group (P > 0.05) (Fig 6A). Moreover, the T2 values in the right medial femoral of the model group were significantly higher than those in the left medial femoral (P < 0.001) (Fig 6B).

## STE results

Elastic modulus values of medial femorals were examined, and the elastic modulus values of the ROI in the bilateral hindlimb medial femorals of the three groups were obtained (Fig 7, S1 Raw Image). The results revealed a more uniform distribution

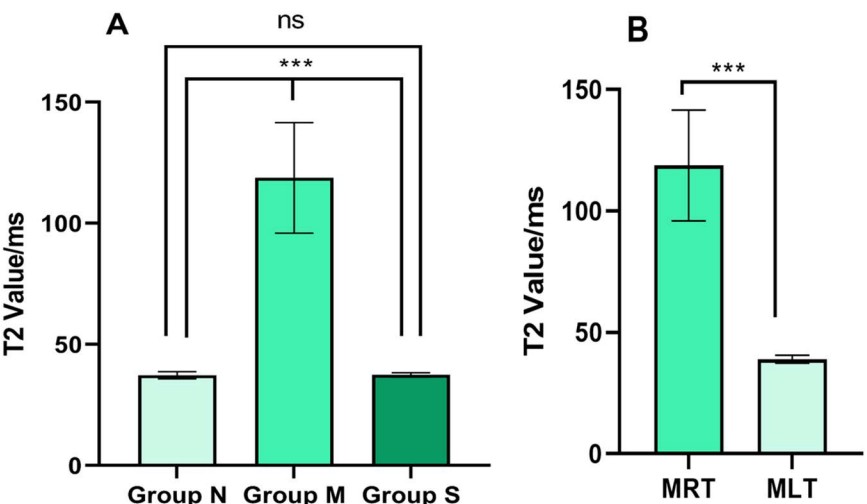

**Fig 6. Comparison of T2 value of medial femoral muscle (n = 5).** A: Comparison of T2 value of right medialis muscle in each group. B: Comparison of T2 value of bilateral medial femoris muscle in model group.***P < 0.001; ns P > 0.05. N: control group; M: model group; S: treatment group; MRT,right medial femoral muscle of model group; MLT,left medial femoral muscle of model group.

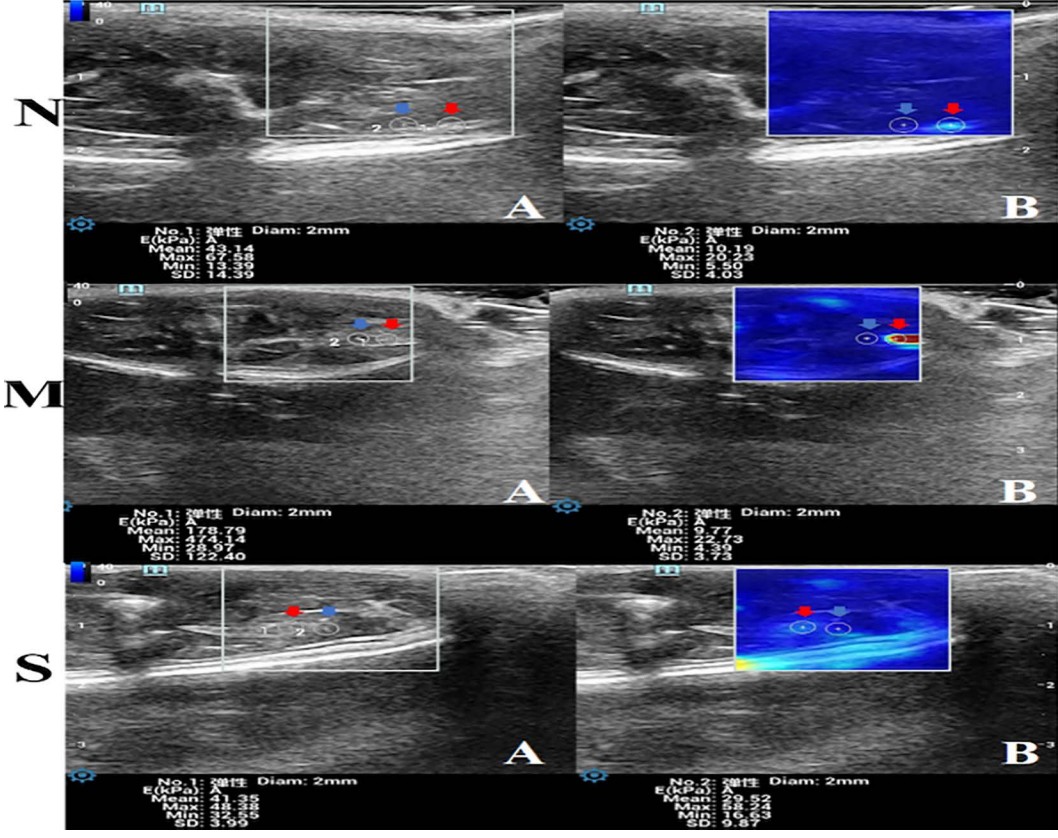

**Fig 7. Sound touch elastography of the right medial femoral muscle (n = 5). A:** gray-scale ultrasound image; **B:** Sound touch elasticity image; **1:** The red arrows indicate TrP areas; **2:** The blue arrows indicate the right TrP adjacent area; N: control group; M: model group; S: treatment group.

and lower modulus values in the control group. The elastic modulus values were relatively low and showed no significant differences between the "TrP area" and the adjacent area (P > 0.05). Comparison of the elastic modulus of the right "TrP area" in the three groups of rats revealed a higher modulus value in the model group than in the control group(P < 0.001). The treatment group showed a lower elastic modulus value than the model group (P < 0.001) but remained higher than the control group (P < 0.01). Furthermore, the mean modulus value of elastic modulus in the right TrP area of the model group was significantly higher than in the adjacent area (P < 0.001) (Fig 8A). The mean modulus value within the Trp area of the right medial femoral was significantly higher than that of the left medial femoral in the model group (P < 0.001) (Fig 8B).

### TEM result

The control group exhibited normal mitochondrial structure, while the model group displayed swollen mitochondria with broken or disappeared cristae and reduced lamella, accompanied by local focal dissolution. In the treatment group, mitochondrial swelling was significantly relieved or tended to normal, with a significant increase in lamellar bodies (Fig 9). 5 rats were tested in each group.

### Western blot results

SIRT3 expression levels in muscle tissues of the three groups of rats are shown in Fig 10A. We observed a significant downregulation of SIRT3 expression in the model group compared with the control group (P < 0.01). After silver needle

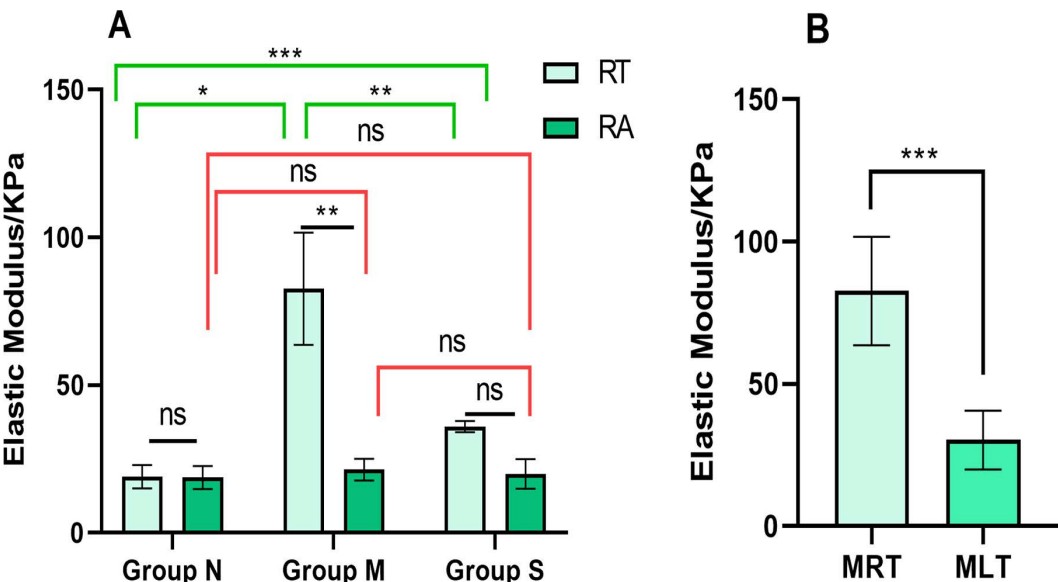

**Fig 8. Comparison of elastic modulus value in ROI of medial femoral muscle (n = 5). A:** Comparison of ROI elastic modulus of right medialis femoral muscle in each group. **B:** Comparison of ROI elastic modulus of bilateral TrP area in model group. ***$P < 0.001$, **$P < 0.01$, *$P < 0.05$, ns $P > 0.05$. N: control group; M: model group; S: treatment group; RT, right TrP area; RA, right TrP adjacent area; MRT, right TrP area of model group; MLT, left TrP area of model group.

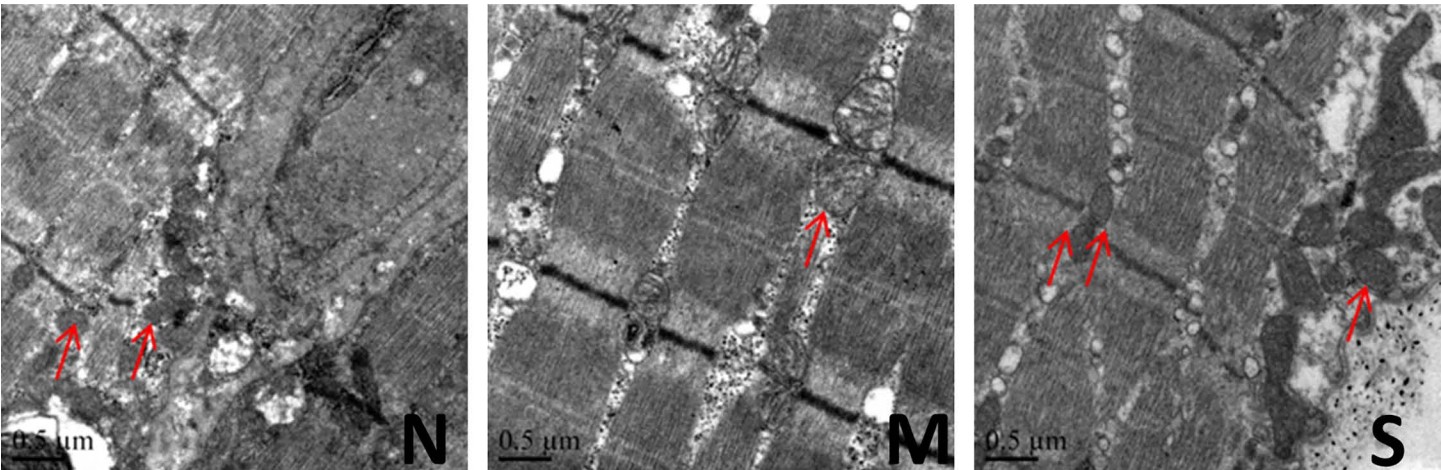

**Fig 9. Right medial femoral muscle TEM sections (n = 5).** Scale bar = 0.5µm. Red arrows point to mitochondria. TEM: transmission electron microscopy; N: control group; M: model group; S: treatment group.

thermal conduction therapy, SIRT3 expression in muscle tissue was significantly upregulated in the treatment group compared with the model group (P < 0.01), with no statistically significant difference between the treatment group and the control group (P > 0.05) (Fig 10B).

## Discussion

In previous studies, there was a lack of quantitative research on the effect of silver needle thermal conduction therapy on muscle changes in TrPs. This study applies T2 mapping and STE techniques to evaluate muscle changes in the TrPs the

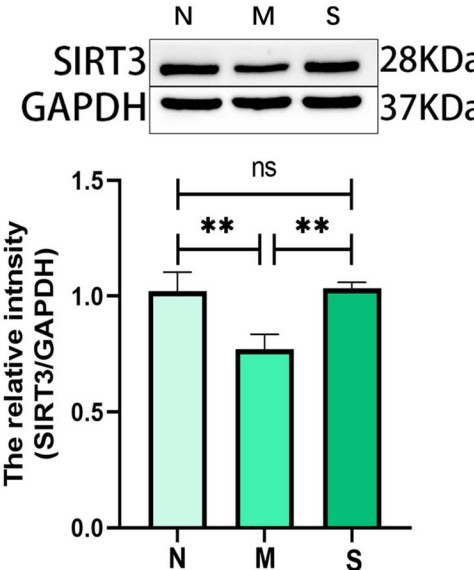

**Fig 10. Comparison of SIRT3 in TrP of medial femoral muscle (n = 5).** A: Western Blot bands for SIRT3 (n = 5). B: Quantitative analysis of SIRT3; **$P < 0.01$, ns $P > 0.05$. TrP: trigger point; N: control group; M: model group; S: treatment group.

reparative effect of silver needle thermal conduction therapy on the TrPs. Additionally, the ultrastructure and function of mitochondria were observed by TEM and Western Blotting to understand the impact of silver needle thermal conduction therapy on mitochondria in TrPs region.

The damage and disarrangement of muscle fibers and SEA in TrPs region of MPS rats indicated that the MPS model was successfully replicated. Silver needle thermal conduction therapy improved the disordered shape and arrangement of muscle fibers. At the same time, it reduced the frequency and amplitude of SEA in MPS rats.The change of pain threshold also verified the above conclusion.

T2 mapping has been shown to be a sensitive tool for evaluating TrPs in rat models [11]. T2 mapping illustrates the distribution of T2 relaxation times (T2 values) in the field of view. The T2 value of normal muscle tissue is relatively short, and it is generally prolonged when inflammation occurs in the tissue. Our study showed a significant increase in T2 values at TrPs in MPS rats compared to normal control rats. silver needle thermal conduction therapy can reduce the T2 value of the right medial femoris muscle in rats. silver needle thermal conduction therapy has a thermal effect. After heating, blood vessels dilate locally in soft tissues, accelerate local blood circulation, and accelerate the metabolism of inflammatory substances. Silver needle thermal conduction therapy may reduce T2 values by reducing the inflammatory response.

STE has gained increasing clinical attention due to its ability to quantitatively analyze tissue stiffness. The higher the modulus value, the higher the object's stiffness and the lower its elasticity. Mindray Resona9's STE provides real-time information on tissue stiffness within the ROI. Our results showed significantly higher muscle stiffness (elastic modulus) at TrPs of MPS rats than that of the control group. Combined with the EMG results that increased spontaneous electrical activity was observed in MPS rats. Muscle depolarization caused muscle contraction, and long-term depolarization caused an energy crisis, eventually leading to muscle contracture. The muscle stiffness decreased significantly after silver needle thermal conduction therapy. We believe that silver needle thermal conduction therapy has the effect of relaxing muscle. Electromyography showed that silver needle thermal conduction therapy reduced the frequency of spontaneous electrical activity in MPS rats. It is suggested that silver needle thermal conduction therapy may destroy TrPs through acupuncture effect and thermal effect, lead to reduction of spontaneous electrical activity, improve muscle contracture, and play a therapeutic role.

The energy crisis theory explains the formation mechanism of TrPs. As the principal energy-producing organelles of the cell, mitochondria support numerous biological processes related to metabolism, growth, and regeneration in skeletal muscle [14]. Persistent inhibition of mitochondrial biogenesis and impaired mitochondrial function can sustain ATP production at a relatively lower level, leading to a persistent energy crisis and ultimately the formation of Trps. Research suggests that stimulating or enhancing mitochondrial biogenesis may represent a novel strategy for MPS treatment [15]. SIRT3 plays a key role in maintaining normal mitochondrial functions. It is characterized by elevated metabolic rates, oxygen consumption rates, and mitochondrial densities [16]. Western blot analysis in our study discerned a significant down-regulation of SIRT3 expression in MPS rats compared with the control group. Concurrently, western blot results suggested that silver needle thermal conduction therapy upregulated SIRT3 expression in the muscle at the TrPs. Meanwhile, we observed the ultrastructure of muscle mitochondria at the TrPs of MPS rats using transmission electron microscopy. Mitochondrial morphology and function are closely related. It is found that at the TrPs of MPS rats, the mitochondria of muscle tissue were swelled, the cristae were broken or disappeared, and the lamella was decreased with local focal lysis. While, silver needle thermal conduction therapy significantly reduced the swelling of muscle mitochondria at the TrPs and significantly increased the lamellar body. These results suggest that silver needle thermal conduction therapy may improve pain in MPS rats by repairing mitochondria.

Silver needle thermal conduction therapy can improve the muscle stiffness in MPS rats. It provides an objective evaluation index for the treatment of MPS with silver needle thermal conduction therapy. Silver needle thermal conduction therapy is a different treatment method from dry needling therapy. Dry needle therapy involves needling TrPs and repeatedly lifting and inserting the needle body to destroy TrPs, or converting active TrPs into potential ones. The silver needle thermal conduction therapy involves densely distributing needles according to the anatomical structure of the painful area. Stop inserting the needle when the tip reaches the periosteum and then heat the needle body. There is no need to repeatedly lift and insert the needle body, no pursuit of precise positioning of TrPs, nor is there a need to induce convulsive reactions.

Literature reports that acupuncture mechanically provides local stretching for shortened muscle segments and contracted cytoskeletal structures. This will restore the muscle segments to their resting length by reducing the overlap degree between actin and myosin filaments [17–19]. The silver needle thermal conduction therapy has a needling effect and may exert therapeutic effects by relaxing muscles. Meanwhile, the deep needle technique of dry needles has been proven to be more effective than the shallow needle technique in treating pain related to myofascial trigger points [20]. The depth of the silver needle thermal conduction therapy reaching the periosteum may be one of the mechanisms by which it exerts good therapeutic effects. At the same time, the silver needle thermal conduction therapy may exert a thermal effect by conducting temperature to the body. Previous literatures has reported that hyperthermia can promote muscle relaxation, enhance blood circulation, and regulate nociceptors [21]. An increased muscle temperature can initially cause substantial improvements in force production, faster rates of force generation, relaxation, shortening, and production of power output [22]. Moxibustion heating can dilate blood vessels, increase blood flow and improve microcirculation [23]. The above results all suggest that the thermal effect plays a role in improving muscle spasms. Compared with other metals, silver has a better thermal conduction effect [24]. The silver needle contains 85% silver. It can better exert the thermal effect. Combined with the result that the silver needle thermal conduction therapy can improve muscle stiffness, we believe that the silver needle thermal conduction therapy may have both mechanical and thermal effects simultaneously. Its specific molecular mechanism needs to be further explored.

TrPs dry needling is most effective to release myofascial pain after local twitch response (LTR) is elicited. The number of sessions for dry needling therapy is not fixed, it can be 1 session or multiple sessions [5]. Bubnov R V pointed out that ultrasound-guided dry needling (US-DN) significantly increased the pain relief effect, increases the level of eliciting LTR, and significantly decreased the average number of needled trigger points and the average number of treatment sessions [25]. DN under US guidance increases treatment effectiveness and reduces complications [26]. The short-term and long-term effects of the two treatment methods still require further comparative studies.

Although silver needles have excellent thermal conductivity, there is no need to worry about side effects such as burns caused by heating, nor about the cost of silver needles. Because the implementation of needle thermal conduction therapy is equipped with heating devices, which have the function of temperature regulation. We control the temperature of the skin injection point of the silver needle at 42°C to ensure the safety of this therapy. Besides, silver has antibacterial properties. Silver ions can form strong bonding bonds with molecules containing oxygen, sulfur, nitrogen and other elements that microorganisms use for respiration, preventing these molecules from being utilized by microorganisms and thus leading to their death. The antibacterial property of silver has enabled it to be widely used in medical and daily necessities. In modern medicine, silver is used to make antibacterial dressings, antibacterial medical devices, etc. Silver is also added to products such as gels and bandages to prevent wound infections. Because silver has a relatively low hardness, it is usually made into alloys by adding other metals to enhance its hardness and wear resistance. The silver needles we use are made of an alloy with an 85% silver content. During many years of clinical application, complications of infection have been very rare. Although the input cost of silver needles is relatively high, they can be reused after disinfection.

However, whether the improvement of mitochondrial function can directly reduce pain needs further proof. At the same time, this experiment did not adopt more detection methods to confirm mitochondrial repair, such as ROS and ATP detection. This experiment did not set up the needling without heating group and the silver-plated needle thermal conduction treatment group to distinguish the effect of metal silver, the needling effect and the thermal effect. The small sample size and lack of long-term efficacy monitoring was also a limitation of the study.

## Conclusion

The silver needle thermal conduction therapy demonstrates the ability to reduce muscle inflammation and stiffness and facilitate the repair of damaged muscle mitochondria. However, it remains to be proven whether the silver needle thermal conduction therapy works through the heat effect alone, the acupuncture effect alone, or both combined.

## Supporting information

**S1 Raw images.** A file containing all the original spots and gel images of the manuscript main figure. (ZIP)

## Author contributions

**Formal analysis:** Zilong Yu.

**Funding acquisition:** Yue Qin, Yuanxin Huang.

**Investigation:** Chunxin Wo.

**Methodology:** Yue Wang, Zilong Yu.

**Project administration:** Yuanxin Huang.

**Software:** Yue Wang, Zhenmin Wang.

**Supervision:** Lin Wang.

**Writing – original draft:** Yue Qin.

**Writing – review & editing:** Lin Wang.

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
