## [Decision Letter · Decision Letter 0]

10 Dec 2024

PONE-D-24-35346Efficacy of Silver Needle Thermal Conduction Therapy on Myofascial Pain Syndrome Rats: Based on Quantitative High-resolution Magnetic Resonance T2 Imaging, Shear Wave Elastrography and Muscle Mitochondrial Ultrastructure.PLOS ONE

Dear Dr. Qin,

Thank you for submitting your manuscript to PLOS ONE. After careful consideration, we feel that it has merit but does not fully meet PLOS ONE’s publication criteria as it currently stands. Therefore, we invite you to submit a revised version of the manuscript that addresses the points raised during the review process.

We look forward to receiving your revised manuscript.

*Comments from PLOS Editorial Office: We note that one or more reviewers has recommended that you cite specific previously published works. As always, we recommend that you please review and evaluate the requested works to determine whether they are relevant and should be cited. It is not a requirement to cite these works. We appreciate your attention to this request.*

Kind regards,

Jiajia Ye

Guest Editor

PLOS ONE

“the National Natural Science Foundation of China (Grants No: 82160226/H2902) , the Natural Science Foundation of China (Grants No: 82060811/H2902) , Guizhou Province Science and Technology Plan Project (Grants No:Qianke He Foundation -ZK[2021] General 508) , Guizhou Province Science and Technology Plan Project (Grants No:Qianke He Foundation -ZK[2023]General 370),  Guizhou Administration of Traditional Chinese Medcine (Grants No: QZYY-2021-123) and Guizhou Province colleges and universities youth science and technology talent development project Grants No:Qianjiao He Foundation -KY[2022] 235�”

6. Please ensure that you include a title page within your main document. We do appreciate that you have a title page document uploaded as a separate file, however, as per our author guidelines (http://journals.plos.org/plosone/s/submission-guidelines#loc-title-page) we do require this to be part of the manuscript file itself and not uploaded separately.

**Comments to the Author**

1. Is the manuscript technically sound, and do the data support the conclusions?

Reviewer #1: Yes

Reviewer #2: Yes

2. Has the statistical analysis been performed appropriately and rigorously? 

Reviewer #1: I Don't Know

Reviewer #2: Yes

3. Have the authors made all data underlying the findings in their manuscript fully available?

Reviewer #1: Yes

Reviewer #2: Yes

4. Is the manuscript presented in an intelligible fashion and written in standard English?

Reviewer #1: Yes

Reviewer #2: Yes

5. Review Comments to the Author

Please use the space provided to explain your answers to the questions above. You may also include additional comments for the author, including concerns about dual publication, research ethics, or publication ethics. 

Reviewer #1: Dear Authors,

Thank you very much for your article titled "Efficacy of Silver Needle Thermal Conduction Therapy on Myofascial Pain Syndrome Rats: Based on Quantitative High-resolution Magnetic Resonance T2 Imaging, Shear Wave Elastrography and Muscle Mitochondrial Ultrastructure". I found your research very interesting and relevant, and I believe it has the potential to make a significant contribution to the understanding of myofascial pain syndrome and its treatment.

However, I have a few observations and suggestions that I hope will help you improve the clarity and rigor of your manuscript.

Points for Clarification:

Inconsistent Rat Numbers: You mention using 24 rats, divided into three groups of 5. However, the HE staining and TEM analyses mention 6 rats per group. Please standardize the number of rats per group for all analyses.

MPS Model: The description of the MPS induction method should be more detailed. Please specify the frequency and duration of the impacts, as well as the type and duration of the physical training imposed on the rats.

Temperature Justification: You indicate that the needle temperature is 110°C, but that the skin surface temperature is 42°C. Explain the choice of these temperatures and how the skin temperature was controlled and maintained constant.

Suggested Improvements:

SWE Analysis: Please specify the size and shape of the ROIs used for the SWE analysis, as well as the method of selecting these ROIs.

SIRT3 Interpretation: The increase in SIRT3 expression alone is not enough to conclude mitochondrial repair. Additional analyses of mitochondrial function would be necessary to support this conclusion.

Limitations: It would be wise to include a section in the discussion to address the limitations of the study, including the small sample size and the lack of confirmation of mitochondrial repair.

Reviewer #2: This manuscript explores the effects of silver needle thermal conduction therapy on myofascial trigger points (TrPs) using advanced imaging modalities like T2 mapping and strain elastography (STE), complemented by mitochondrial evaluations. The study provides intriguing data and demonstrates high scientific value. However, several critical points need to be addressed to enhance its rigor and clarity.

Strengths

Sophisticated Design: The use of high-resolution magnetic resonance imaging (T2 mapping) and STE to quantitatively assess TrPs is commendable.

Innovative Approach: Employing silver needle thermal conduction therapy and evaluating its effects on mitochondrial structure and function is novel.

Comprehensive Analysis: Combining structural (imaging) and functional (mitochondrial) parameters adds depth to the study.

Major Concerns

Targeting Accuracy of TrPs:

The study does not clearly address whether TrPs were precisely targeted. Imaging-based guidance (e.g., ultrasound) should be detailed to verify accurate targeting before and after the therapy.

Referencing established protocols for TrP detection using ultrasound, such as those by Bubnov et al., would improve the methodological section [Bubnov R.V. Evidence-based pain management: is the concept of integrative medicine applicable?. EPMA Journal 3, 13 (2012). https://doi.org/10.1186/1878-5085-3-13

Bubnov RV: The use of trigger point dry needling under ultrasound guidance for the treatment of myofascial pain (technological innovation and literature review). Lik Sprava. 2010, 5–6: 56-64.]

US Imaging and Needling:

Ultrasound images should be included to showcase how TrPs were detected and evaluated pre- and post-treatment.

Dry needling, widely recognized as effective for TrPs, should be compared to silver needle thermal conduction therapy. This will establish the added value of the thermal conduction method.

Mechanism of Action:

The manuscript does not adequately explain why silver needles were chosen. What unique mechanisms (chemical, mechanical, or thermal) contribute to their effects?

The absence of discussion regarding local twitch responses evoked during treatment is a missed opportunity to connect EMG findings with imaging data.

Title and Focus:

The title should explicitly reflect needling for TrPs instead of the generalized term "Myofascial Pain Syndrome."

Consider shortening the title to improve clarity and focus.

Visualization:

The images do not sufficiently reflect TrPs as a target. Highlighting the precise region of TrPs in T2 mapping or STE images is essential.

Clear visualization would strengthen the study's impact and reproducibility.

Mitochondrial Analysis:

While the mitochondrial findings are compelling, their relevance to the clinical outcomes should be elaborated. Does repairing mitochondrial damage directly correlate with pain relief or muscle function restoration?

Recommendations

Methods: Include a detailed description of ultrasound imaging protocols for TrP detection with references, and explain pre- and post-evaluation methods.

Comparative Analysis: Provide a robust comparison of silver needle therapy with dry needling or other mechanical stimulation methods.

Mechanisms: Discuss the rationale for silver needles and clarify the dominant mechanism (thermal vs. mechanical).

Images: Add annotated imaging of TrPs for better visualization and reproducibility.

Title Revision: Refine the title to emphasize needling for TrPs and its therapeutic implications

6. PLOS authors have the option to publish the peer review history of their article (what does this mean? ). If published, this will include your full peer review and any attached files.

---

## [Author Response · Author response to Decision Letter 1]

13 Feb 2025

We feel great thanks for your professional review work on our article. As you are concerned, there are several problems that need to be addressed. According to your nice suggestions, we have made extensive corrections to our previous draft, the detailed corrections are listed below.

To Reviewer #1: 

Points for Clarification:

Inconsistent Rat Numbers: You mention using 24 rats, divided into three groups of 5. However, the HE staining and TEM analyses mention 6 rats per group. Please standardize the number of rats per group for all analyses.

Reply: We were really sorry for our careless mistakes. We have made corrections as requested,as seen in page 5, line 93.

MPS Model: The description of the MPS induction method should be more detailed. Please specify the frequency and duration of the impacts, as well as the type and duration of the physical training imposed on the rats.

Reply: Thank you very much for your reminding. We have made corrections as requested,as seen in page 6, line 121 and 125.

Temperature Justification: You indicate that the needle temperature is 110°C, but that the skin surface temperature is 42°C. Explain the choice of these temperatures and how the skin temperature was controlled and maintained constant.

Reply: I apologize for causing confusion to you. Previous studies believe that the temperature of the silver needle body at the skin injection point should not exceed 42 ° C from the perspective of safety and effectiveness. Silver needle heating instrument conducts heat to the needle body by connecting with the tail end of the needle body. The temperature of the silver needle heating instrument is set to 110° C. After heat dissipation through the exposed needle body, silver needle body at the skin injection point does not exceed 42°C. It has been revised in manuscript, as seen in page 7,line 131.

Suggested Improvements:

SWE Analysis: Please specify the size and shape of the ROIs used for the SWE analysis, as well as the method of selecting these ROIs.

Reply: In the revised manuscript, the size and shape of the ROIs has been organized. It can be located on page 9�line 181. And we have described in more detail the methods for selecting these ROIs. The select method and the parameters of ROI are based on previous references Lv H, Li Z, Hu T, Wang Y, Wu J, Li Y. The shear wave elastic modulus and the increased nuclear factor kappa B (NF-kB/p65) and cyclooxygenase-2 (COX-2) expression in the area of myofascial trigger points activated in a rat model by blunt trauma to the vastus medialis. J Biomech. 2018 Jan 3;66:44-50. The method of selecting these ROIs.

SIRT3 Interpretation: The increase in SIRT3 expression alone is not enough to conclude mitochondrial repair. Additional analyses of mitochondrial function would be necessary to support this conclusion.

Reply We concur with you that more mitochondrial function results would be benefical.In this study, mitochondrial microstructure was used to evaluate mitochondrial function. We have re-written this this part. In the revised manuscript, we combined SIRT3 with mitochondrial microstructure to evaluate mitochondrial repair�as seen in page 19, line 310-324. Meanwhile, we have revised the manuscript as your recommendation.We have added a section in the discussion explaining the limitations of this paper�including the small sample size and the lack of confirmation of mitochondrial repair.Looking ahead, we will pay more attention to enriching experimental methods, such as increasing the detection of ROS, ATP and other indicators.

Limitations: It would be wise to include a section in the discussion to address the limitations of the study, including the small sample size and the lack of confirmation of mitochondrial repair.

Reply We think this is an excellent suggestion. We have added this section to the last paragraph of the discussion�as seen in page 20 ,line 327-329.

To Reviewer #2:

Major Concerns

Targeting Accuracy of TrPs:

The study does not clearly address whether TrPs were precisely targeted. Imaging-based guidance (e.g., ultrasound) should be detailed to verify accurate targeting before and after the therapy.

Referencing established protocols for TrP detection using ultrasound, such as those by Bubnov et al., would improve the methodological section [Bubnov R.V. Evidence-based pain management: is the concept of integrative medicine applicable?. EPMA Journal 3, 13 (2012). https://doi.org/10.1186/1878-5085-3-13

Bubnov RV: The use of trigger point dry needling under ultrasound guidance for the treatment of myofascial pain (technological innovation and literature review). Lik Sprava. 2010, 5–6: 56-64.]

Reply: We sincerely appreciate your valuable comments. Silver needle heat conduction therapy has been used in clinical practice for decades, and it has good long-term curative effect. However, the evaluation criteria are subjective, mainly through VAS score. More objective evidence is lacking. Therefore, the purpose of this study was to quantitatively evaluate the efficacy of silver needle thermal conductivity therapy by T2mapping and STE, and to provide stronger evidence support for the treatment of MPS by this technology. In clinical practice, we determine the acupuncture site on the basis of anatomy by palpation of tension band and muscle twitching caused by acupuncture. Therefore, in order to avoid the influence of different manipulation methods on the results, we locate TrP by palpation combined with electromyography induced muscle twitching in animal experiments, without the use of image guidance for localization. We have refined and modified this content,which can be seen in page 8, line 174. Imaging is only used to assess the effectiveness of treatment. Your suggestions have provided us with a more optimized treatment plan, and we will carry out more accurate treatment according to your suggestions in the later stage, which may achieve more therapeutic effects and less pain.

US Imaging and Needling:

Ultrasound images should be included to showcase how TrPs were detected and evaluated pre- and post-treatment.

Dry needling, widely recognized as effective for TrPs, should be compared to silver needle thermal conduction therapy. This will establish the added value of the thermal conduction method.

Reply:We sincerely thanks for your careful reading and suggestion.We concur with you that more dry needling comparison would be benefical. However, due to the long modeling time (3 months) and the need to synchronize the existing three groups of animals which lead to animals waste, it is impractical to conduct additional experiments at present. On the other hand, the main purpose of this experiment is to conduct an quantitative evaluation of the curative effect of silver needle thermal conduction therapy, in order to provide more objective evidence for the silver needle thermal conduction treatment of MPS and a new treatment for MPS. Therefore, comparative studies with other treatments were not planned at the time of design. Looking ahead, we will put more emphasis on the efficacy of different treatments.

Mechanism of Action:

The manuscript does not adequately explain why silver needles were chosen. What unique mechanisms (chemical, mechanical, or thermal) contribute to their effects?

The absence of discussion regarding local twitch responses evoked during treatment is a missed opportunity to connect EMG findings with imaging data.

Reply:We think this is an excellent suggestion. In the revised manuscript, our explanation and discussion have been added,which can be found in page 3,line 45-49 and page19, line 302-309.

Title and Focus:

The title should explicitly reflect needling for TrPs instead of the generalized term "Myofascial Pain Syndrome."

Consider shortening the title to improve clarity and focus.

Reply: We feel great thanks for your professional review work on our article. We have revised our tittle.

Visualization:

The images do not sufficiently reflect TrPs as a target. Highlighting the precise region of TrPs in T2 mapping or STE images is essential.

Clear visualization would strengthen the study's impact and reproducibility.

Reply: We first palpate the muscle tension band and then identify the precise location of spontaneous electrical activity detected by EMG as TrP. Mark the location before performing a follow-up check.We revised it on the 174 line of page 8.

Mitochondrial Analysis:

While the mitochondrial findings are compelling, their relevance to the clinical outcomes should be elaborated. Does repairing mitochondrial damage directly correlate with pain relief or muscle function restoration?

Reply: According to your suggestion, we explained the relationship between Trp production and mitochondrial dysfunction in more detail, so as to conclude that silver needle heat conduction therapy may play a role in relieving pain by repairing mitochondria( page 19,line 310-324). However, it has not been proven that the functional repair of mitochondria can directly improve MPS pain. This is also the limitation of this experiment. Therefore, we have added this section to the last paragraph of the discussion,page 20, line 327-329.

Recommendations

Methods: Include a detailed description of ultrasound imaging protocols for TrP detection with references, and explain pre- and post-evaluation methods.

Reply: In the revised manuscript, the size and shape of the ROIs has been organized. And we have described in more detail the methods for selecting these ROIs, which can be seen in page 8, line 174 and page 9, line 181-184.

Comparative Analysis: Provide a robust comparison of silver needle therapy with dry needling or other mechanical stimulation methods.

Reply: Thank you for your suggestion. Your suggestion can more clearly explain the thermal effect of silver needle conduction therapy. For reasons such as time and animal waste, it is difficult for us to add this part in this article. In future studies, we will pay more attention to the comparison between different treatments.

Mechanisms: Discuss the rationale for silver needles and clarify the dominant mechanism (thermal vs. mechanical).

Reply:We have revised our manuscript, which can be seen in page 20, line 295-298,302-309.

Images: Add annotated imaging of TrPs for better visualization and reproducibility.

Reply: We re-labeled the TrP according to the comments.

Title Revision: Refine the title to emphasize needling for TrPs and its therapeutic implications

Reply: We have revised our title, which can be found in page 1, line 1.

---

## [Decision Letter · Decision Letter 1]

7 Mar 2025

PONE-D-24-35346R1Quantitative evaluation of therapeutic effect of silver needle thermal conduction therapy on myofascial trigger pointPLOS ONE

Dear Dr. Qin,

Thank you for submitting your manuscript to PLOS ONE. After careful consideration, we feel that it has merit but does not fully meet PLOS ONE’s publication criteria as it currently stands. Therefore, we invite you to submit a revised version of the manuscript that addresses the points raised during the review process.

We look forward to receiving your revised manuscript.

Kind regards,

Jiajia Ye

Guest Editor

PLOS ONE

Journal Requirements:

**Additional Editor Comments:**

The manuscript has been significantly improved since the initial submission, with clearer explanations and refined methodology. However, a major conceptual issue remains unresolved: the justification for selecting silver needle thermal conduction therapy (SNTCT) over conventional dry needling (DN). This aspect must be explicitly addressed throughout the manuscript to enhance its scientific rigor and clinical relevance.

Major Issues:

1. Justification for Silver Needle Thermal Conduction Therapy (SNTCT)

The manuscript should clearly define why the authors chose this method over conventional DN. The following aspects require detailed discussion:

Why introduce silver and heat to the needle when standard fine needles are already effective for trigger point inactivation?

What specific physiological or biomechanical advantages does silver provide in this context? If its antimicrobial properties or conductivity are relevant, these should be supported with evidence.

How does thermal conduction enhance the therapeutic effect beyond what is achieved with precise DN?

Is there prior research supporting the addition of these variables in needling therapy? If not, a stronger theoretical basis should be provided.

If silver is only used as a thermal conductor, why not use another material?

2. Mechanism of Action & Physiological Effects

The study should clarify the hypothesized mechanisms through which silver and heat alter the effects of DN:

The standard effectiveness of DN is largely attributed to precise intervention and eliciting localized twitch responses (LTRs) in trigger points. How do heat and silver modify this response?

Could thermal conduction alter needle sensitivity, thereby influencing tactile feedback and the ability to localize trigger points?

If heat increases circulation or pain modulation, was this measured or supported by prior research?

Were adverse effects (e.g., risk of burns, altered tissue response) considered?

3. Study Design Considerations

To isolate the true effect of silver and heat, the study should ideally include:

A comparison between standard DN, silver-coated DN, and silver-thermal DN to assess individual contributions.

A discussion on whether blinding was possible and how placebo effects were minimized.

If no direct comparison was made, the authors should acknowledge this as a limitation and suggest future studies to address it.

4. Clinical Applicability & Cost-Benefit Considerations

If precise DN is already effective, does the added complexity and potential cost of SNTCT justify its use?

Are there specific clinical scenarios where silver thermal conduction needling would be superior to traditional DN?

Could this method introduce risks or complications that are not present with standard DN?

5. Needling Precision & Local Twitch Responses (LTRs)

The therapeutic effectiveness of DN is highly dependent on precise needling technique and the elicitation of LTRs. The manuscript does not discuss whether silver-thermal needling affects this precision.

The authors should provide detailed descriptions of their needling protocol to ensure consistency with established DN methodologies, such as those described in [https://pmc.ncbi.nlm.nih.gov/articles/PMC3533862/].

If LTRs were monitored, this should be explicitly stated. If not, the authors should acknowledge that this is a limitation of the study.

Minor Issues:

Terminology consistency: Ensure that all references to the intervention (SNTCT, silver-thermal DN, etc.) are consistent throughout the manuscript.

Literature review expansion: Include more references to existing studies comparing different needling techniques.

Clarify study limitations: If the study does not separate silver and heat effects, this should be acknowledged.

Conclusion:

The manuscript must provide a strong, evidence-based justification for incorporating silver and heat into DN. Additionally, the study design should either control for these variables separately or acknowledge this as a limitation. Addressing these concerns will significantly enhance the manuscript’s scientific and clinical relevance.

Reviewers' comments:

Reviewer's Responses to Questions

**Comments to the Author**

1. If the authors have adequately addressed your comments raised in a previous round of review and you feel that this manuscript is now acceptable for publication, you may indicate that here to bypass the “Comments to the Author” section, enter your conflict of interest statement in the “Confidential to Editor” section, and submit your "Accept" recommendation.

Reviewer #1: All comments have been addressed

Reviewer #2: All comments have been addressed

2. Is the manuscript technically sound, and do the data support the conclusions?

Reviewer #1: Yes

Reviewer #2: Yes

3. Has the statistical analysis been performed appropriately and rigorously? 

Reviewer #1: Yes

Reviewer #2: Yes

4. Have the authors made all data underlying the findings in their manuscript fully available?

Reviewer #1: Yes

Reviewer #2: Yes

5. Is the manuscript presented in an intelligible fashion and written in standard English?

Reviewer #1: Yes

Reviewer #2: Yes

6. Review Comments to the Author

Reviewer #1: Dear Author

I am writing to express my sincere gratitude for the modifications you have made to your manuscript. Your diligence in addressing the concerns raised by the reviewers has significantly improved the precision and accuracy of the article.

Reviewer #2: The manuscript has been significantly improved since the initial submission, with clearer explanations and refined methodology. However, a major conceptual issue remains unresolved: the justification for selecting silver needle thermal conduction therapy (SNTCT) over conventional dry needling (DN). This aspect must be explicitly addressed throughout the manuscript to enhance its scientific rigor and clinical relevance.

Major Issues:

1. Justification for Silver Needle Thermal Conduction Therapy (SNTCT)

The manuscript should clearly define why the authors chose this method over conventional DN. The following aspects require detailed discussion:

Why introduce silver and heat to the needle when standard fine needles are already effective for trigger point inactivation?

What specific physiological or biomechanical advantages does silver provide in this context? If its antimicrobial properties or conductivity are relevant, these should be supported with evidence.

How does thermal conduction enhance the therapeutic effect beyond what is achieved with precise DN?

Is there prior research supporting the addition of these variables in needling therapy? If not, a stronger theoretical basis should be provided.

If silver is only used as a thermal conductor, why not use another material?

2. Mechanism of Action & Physiological Effects

The study should clarify the hypothesized mechanisms through which silver and heat alter the effects of DN:

The standard effectiveness of DN is largely attributed to precise intervention and eliciting localized twitch responses (LTRs) in trigger points. How do heat and silver modify this response?

Could thermal conduction alter needle sensitivity, thereby influencing tactile feedback and the ability to localize trigger points?

If heat increases circulation or pain modulation, was this measured or supported by prior research?

Were adverse effects (e.g., risk of burns, altered tissue response) considered?

3. Study Design Considerations

To isolate the true effect of silver and heat, the study should ideally include:

A comparison between standard DN, silver-coated DN, and silver-thermal DN to assess individual contributions.

A discussion on whether blinding was possible and how placebo effects were minimized.

If no direct comparison was made, the authors should acknowledge this as a limitation and suggest future studies to address it.

4. Clinical Applicability & Cost-Benefit Considerations

If precise DN is already effective, does the added complexity and potential cost of SNTCT justify its use?

Are there specific clinical scenarios where silver thermal conduction needling would be superior to traditional DN?

Could this method introduce risks or complications that are not present with standard DN?

5. Needling Precision & Local Twitch Responses (LTRs)

The therapeutic effectiveness of DN is highly dependent on precise needling technique and the elicitation of LTRs. The manuscript does not discuss whether silver-thermal needling affects this precision.

The authors should provide detailed descriptions of their needling protocol to ensure consistency with established DN methodologies, such as those described in [https://pmc.ncbi.nlm.nih.gov/articles/PMC3533862/].

If LTRs were monitored, this should be explicitly stated. If not, the authors should acknowledge that this is a limitation of the study.

Minor Issues:

Terminology consistency: Ensure that all references to the intervention (SNTCT, silver-thermal DN, etc.) are consistent throughout the manuscript.

Literature review expansion: Include more references to existing studies comparing different needling techniques.

Clarify study limitations: If the study does not separate silver and heat effects, this should be acknowledged.

Conclusion:

The manuscript must provide a strong, evidence-based justification for incorporating silver and heat into DN. Additionally, the study design should either control for these variables separately or acknowledge this as a limitation. Addressing these concerns will significantly enhance the manuscript’s scientific and clinical relevance.

7. PLOS authors have the option to publish the peer review history of their article (what does this mean? ). If published, this will include your full peer review and any attached files.

**Do you want your identity to be public for this peer review?** For information about this choice, including consent withdrawal, please see our Privacy Policy .

Reviewer #1: **Yes: ** Dr Paul ELHOMSY

Reviewer #2: **Yes: ** Rostyslav Bubnov

---

## [Author Response · Author response to Decision Letter 2]

20 Apr 2025

Thank you very much for the valuable comments, which have been very helpful for our research and manuscript. We have responded and made modifications to your question, as follows.

Major Issues:

1. Justification for Silver Needle Thermal Conduction Therapy (SNTCT)

The manuscript should clearly define why the authors chose this method over conventional DN. The following aspects require detailed discussion:

Why introduce silver and heat to the needle when standard fine needles are already effective for trigger point inactivation?

What specific physiological or biomechanical advantages does silver provide in this context? If its antimicrobial properties or conductivity are relevant, these should be supported with evidence.

How does thermal conduction enhance the therapeutic effect beyond what is achieved with precise DN?

Is there prior research supporting the addition of these variables in needling therapy? If not, a stronger theoretical basis should be provided.

If silver is only used as a thermal conductor, why not use another material?

Reply Thank you very much for your valuable comments. First of all, silver needles and dry needles are very different: 1, the treatment site is different: silver needles are based on the anatomical structure of the pain site, and the treatment is performed at the starting point, stopping point or shape of the muscle, while dry needles are treated according to the acupuncture point; 2. Different acupuncture depth: the depth of the silver needle reaches the periosteal, and the action site of the dry needle is relatively shallow; 3. Different thickness: the silver needle body is thicker, the diameter is about 1.1mm, the length is between 110-150mm, the dry needle diameter is about 0.3-0.5mm, the length is about 40-50mm [3, 4], the length and diameter of the silver needle is about 3 times that of the dry needle; 4. Different treatment times: silver needle thermal conduction therapy only requires a single treatment, while dry needle therapy usually requires regular multiple treatments, as seen in page 3, line 46.

Secondly, research indicates that dry acupuncture has poor long-term results [16, 17]. For patients experiencing extreme pain, the impact is greatly diminished[18]. The clinical advantages of dry acupuncture in improving functional disorders and its follow-up effects remain unclear [19].When it comes to improving motor function and everyday life activities and reducing spasticity,warm acupuncture is better than electroacupuncture or acupuncture[27]. The clinical effect of fire acupuncture in treating spasticity after stroke is superior to that of traditional acupunctur[28]. The therapeutic effect of intensive moxibustion plus acupuncture is superior to that of simple acupuncture in improving symptoms of frozen shoulder in patients [29]. It is suggested that thermal effect combined with mechanical effect is superior to mechanical effect alone. silver needle thermal conduction therapy has both thermal and mechanical effects, as seen in page 14, line 323 and page 14, line 341.

Silver has superior thermal conductivity compared to other metals, which is why silver needles are utilized[30]. However, because the application of silver needle thermal conduction therapy is equipped with heating equipment that has the function of temperature control, and we control the temperature of the skin injection point of the silver needle below 42℃, there is no need to worry about side effects like scalding caused by heating. The safety of this treatment has been validated by numerous Chinese literatures, as seen in page 14, line 346.

References:

[3]Ziaeifar M, Arab AM, Mosallanezhad Z, Nourbakhsh MR. Dry needling versus trigger point compression of the upper trapezius: a randomized clinical trial with two-week and three-month follow-up. J Man Manip Ther. 2019 Jul;27(3):152-161.

[4]Jo HR, Choi SK, Sung WS, Lee SD, Lee BW, Kim EJ. Thermal Properties of Warm- versus Heated-Needle Acupuncture. Evid Based Complement Alternat Med. 2022 Feb 28;2022:4159172.

[16]Rodríguez-Mansilla J, González-Sánchez B, De Toro García Á, Valera-Donoso E, Garrido-Ardila EM, Jiménez-Palomares M, González López-Arza MV. Effectiveness of dry needling on reducing pain intensity in patients with myofascial pain syndrome: a Meta-analysis. J Tradit Chin Med. 2016 Feb;36(1):1-13.

[17] Gattie E, Cleland JA, Snodgrass S. The Effectiveness of Trigger Point Dry Needling for Musculoskeletal Conditions by Physical Therapists: A Systematic Review and Meta-analysis. J Orthop Sports Phys Ther. 2017 Mar;47(3):133-149. [18]Gerber LH, Sikdar S, Aredo JV, Armstrong K, Rosenberger WF, Shao H, Shah JP. Beneficial Effects of Dry Needling for Treatment of Chronic Myofascial Pain Persist for 6 Weeks After Treatment Completion. PM R. 2017 Feb;9(2):105-112.

[19]Liu L, Huang QM, Liu QG, Thitham N, Li LH, Ma YT, Zhao JM. Evidence for Dry Needling in the Management of Myofascial Trigger Points Associated With Low Back Pain: A Systematic Review and Meta-Analysis. Arch Phys Med Rehabil. 2018 Jan;99(1):144-152.e2.

[27]Yang L, Tan JY, Ma H, Zhao H, Lai J, Chen JX, Suen LKP. Warm-needle moxibustion for spasticity after stroke: A systematic review of randomized controlled trials. Int J Nurs Stud. 2018 Jun;82:129-138.

[28]Qiu X, Gao Y, Zhang Z, Cheng S, Zhang S. Fire Acupuncture versus conventional acupuncture to treat spasticity after stroke: A systematic review and meta-analysis. PLoS One. 2021 Apr 9;16(4):e0249313.

[29]Gao L, Li X, Wang DB, Du ML, Xie J, Gao XY. [Clinical trial of treatment of frozen shoulder by intensive moxibustion plus acupuncture]. Zhen Ci Yan Jiu. 2019 Apr 25;44(4):297-301. Chinese.

[30]Lin L, Cheng K, Shen XY. [Analysis on the temperature-time curve in warm needling manipulation with acupuncture needles of different materials]. Zhongguo Zhen Jiu. 2019 Dec 12;39(12):1301-7. Chinese.

2.Mechanism of Action & Physiological Effects

The study should clarify the hypothesized mechanisms through which silver and heat alter the effects of DN:

The standard effectiveness of DN is largely attributed to precise intervention and eliciting localized twitch responses (LTRs) in trigger points. How do heat and silver modify this response?

Could thermal conduction alter needle sensitivity, thereby influencing tactile feedback and the ability to localize trigger points?

If heat increases circulation or pain modulation, was this measured or supported by prior research?

Were adverse effects (e.g., risk of burns, altered tissue response) considered?

Reply:Thank you very much for your suggestion and we add further clarification to this section.Silver needles may exert therapeutic effects through thermal and mechanical effects. It has been reported in the literature that the needling action mechanically provides a localized stretch to shortened muscle segments and contracted cytoskeletal structures. This would restore the muscle segment to its resting length by reducing the degree of overlap between actin and myosin filaments[20-22]. Silver needle conductive therapy has a needling effect, which may exert its therapeutic effect by relaxing the muscles. Also, deep needling with dry needling has been shown to be more effective than shallow needling in treating pain associated with myofascial trigger points [23]. The depth of action of silver needle thermal conduction therapy reaches the periosteum, which may be one of the mechanisms by which it exerts good therapeutic effects. At the same time, silver needle heat-conducting therapy may exert a thermal effect by transporting temperature to the body. The literature suggests that warm acupuncture is superior to electroacupuncture or moxibustion in reducing spasticity and promoting motor function and activities of daily living. When comparing warm acupuncture to electroacupuncture or moxibustion, the pooled results for spasticity effects and motor function were significant[27]. Compared to traditional acupuncture, fire acupuncture was more clinically beneficial in treating post-stroke spasticity[28].Intensive moxibustion plus acupuncture was superior to simple acupuncture in improving patients' frozen shoulder symptoms [29]. All of the above results suggest that the heat effect plays a role in improving spasticity.Combined with the result that silver needle thermal conduction therapy can improve muscle hardness, we speculate that silver needle thermal conduction therapy may have both mechanical and thermal effects. Its specific mechanism of action needs to be further explored�which can be seen in page14, line 330.

As the needling site, needle depth, and treatment modality (instrument heating, needle retention for 15-20 min, no repeated lifting and insertion, etc.) of silver needle are different from that of dry needle, there is no need to trigger a local twitch response and localize the trigger point during the implementation process. Trigger points are caused by localized muscle contracture, and silver needle thermal conduction therapy may improve muscle spasm through thermal effect, thus exerting analgesic effect instead of simply destroying trigger points directly, as seen in page 13, line 318.

Heat can increase circulation or pain modulation. Previous literature has reported that heat therapy promotes muscle relaxation, enhances blood circulation, and modulates injury receptors [24].An increased muscle temperature can initially cause substantial improvements in force production, faster rates of force generation, relaxation, shortening, and production of power output [25].Moxibustion heating dilates blood vessels, increases blood flow, and improves microcirculation [26]�as seen in page 14, line 336.

The implementation of silver needle thermal therapy is equipped with a heating device that has the function of temperature regulation. We control the temperature of the skin entry point of the silver needle to below 42℃, which does not cause burns or necrosis of the muscle tissue.

Reference:

[20]Dommerholt J. Dry needling in orthopedic physical therapy practice. Orthop Phys Ther Pract. 2004;16 (3):15–20.

[21]Luke DR. Therapeutic needling in osteopathic practice: an evidence-informed perspective. Int J Osteopath Med. 2009;12(1):2–13.

[22]Dommerholt J, Del Moral OM, Gröbli C. Trigger point dry needling. In: Myofascial Trigger Points: pathophysiology and evidence-informed diagnosis and management. Jones & Bartlett Learning, 2009. p. 159–190.

[23]Kalichman L, Vulfsons S. Dry needling in the management of musculoskeletal pain. J Am Board Fam Med. 2010 Sep-Oct;23(5):640-6.

[24]Rossi R. Heat therapy for different knee diseases: expert opinion. Front Rehabil Sci. 2024 Jul 4;5:1390416. doi: 10.3389/fresc.2024.1390416. PMID: 39055174; PMCID: PMC11270809.

3.Study Design Considerations

To isolate the true effect of silver and heat, the study should ideally include:

A comparison between standard DN, silver-coated DN, and silver-thermal DN to assess individual contributions.

A discussion on whether blinding was possible and how placebo effects were minimized.

If no direct comparison was made, the authors should acknowledge this as a limitation and suggest future studies to address it.

Reply: Thank you very much for your valuable comments, which are very important for a deeper comprehension of the mechanisms of silver needle thermal therapy and may even reducing healthcare expenses. The study's shortcomings have been noted and incorporated in the discussion section (as seen in page15, line 365). In the subsequent experiments, we will add groups as per your suggestions to further clarify the thermal effect and needling effect of the silver needle.

4. Clinical Applicability & Cost-Benefit Considerations

If precise DN is already effective, does the added complexity and potential cost of SNTCT justify its use?

Are there specific clinical scenarios where silver thermal conduction needling would be superior to traditional DN?

Could this method introduce risks or complications that are not present with standard DN?

Reply: Long-term effect of dry needling is not good. In our clinical work we have found that silver needle thermal conduction therapy has definite and long term efficacy for MPS patients who are not well treated with dry needling. I myself am a patient of MPS in neck and shoulder, and the pain was relieved briefly after massage, oral medication and dry needling. By chance, I tried silver needle thermal conduction therapy, and after only one treatment, the pain was significantly relieved, and it has been nearly 10 years since then. As a result, I became interested in this folk therapy and conducted preliminary research. At the same time, silver needles can be sterilized and reused repeatedly, and although the input cost is high, it can be used for a long time.

Based on the good long-term efficacy of silver needles and the number of treatments in a single session, I believe that silver needle thermal conduction therapy is especially necessary.

The implementation of silver needle thermal conduction therapy is equipped with a heating device that has the function of temperature regulation, and we control the temperature at the point of entry into the skin of the silver needle to less than 42℃. The safety of this therapy is confirmed by a large number of literatures in our country.

5. Needling Precision & Local Twitch Responses (LTRs)

The therapeutic effectiveness of DN is highly dependent on precise needling technique and the elicitation of LTRs. The manuscript does not discuss whether silver-thermal needling affects this precision.

The authors should provide detailed descriptions of their needling protocol to ensure consistency with established DN methodologies, such as those described in [https://pmc.ncbi.nlm.nih.gov/articles/PMC3533862/].

If LTRs were monitored, this should be explicitly stated. If not, the authors should acknowledge that this is a limitation of the study.

Reply: In fact, the silver needle thermal conduction therapy does not locate based on trigger points, but rather arranges the needles according to the anatomy of the painful area and the tense zone, such as the starting point, ending point or course of the muscle in the painful area. A localized twitch response may be induced during needling. The needling and thermal effects may exert analgesic effect through other pathways than simply destroying the trigger point directly. The fact that it is not easy to precisely localize the trigger point, and it is difficult to implement even under ultrasound guidance, may be one of the reasons why the efficacy of dry needling is unstable and why ultrasound-guided dry needling has not been widely promoted. We describe this in more detail on page 14, line 326. Long-term efficacy monitoring was not performed, which is a shortcoming of this experiment and is described on page 15, line 367.

Minor issue:

Terminology consistency: Ensure that all references to the intervention (SNTCT, silver-thermal DN, etc.) are consistent throughout the manuscript.

Reply We have harmonized this issue.

Literature review expansion: Include more references to existing studies comparing different needling techniques.

Reply The differences between dry needle and silver needle have been explained in more detail in the manuscript�as seen in page13, line 318.

Clarify study limitations: If the study does not separate silver and heat effects, this should be acknowledged.

Reply We recognize this limitation and account for it in the manuscript, as seen in page 15, line 367.

Conclusion:

The manuscript must provide a strong, evidence-based justification for incorporating silver and heat into DN. Additionally, the study design should either control for these variables separately or acknowledge this as a limitation. Addressing these concerns will significantly enhance the manuscript’s scientific and clinical relevance.

Reply Your comments have been very helpful. In the manuscript we did not validate the thermal and pinprick effects of silver needle conductive therapy, and we have explained this limitation. And, we have added this part of the exploration in the new study and hope to report this issue more scientifically in future manuscripts. Thank you very much for your serious questions and scientific advice on the

---

## [Editor Report · Decision Letter 2]

14 May 2025

PONE-D-24-35346R2Quantitative evaluation of therapeutic effect of silver needle thermal conduction therapy on myofascial trigger pointPLOS ONE

Dear Dr. Wang,

Thank you for submitting your manuscript to PLOS ONE. After careful consideration, we feel that it has merit but does not fully meet PLOS ONE’s publication criteria as it currently stands. Therefore, we invite you to submit a revised version of the manuscript that addresses the points raised during the review process.

We look forward to receiving your revised manuscript.

Kind regards,

Jiajia Ye

Guest Editor

PLOS ONE

Journal Requirements:

Additional Editor Comments:

Please address the comments from the reviewers below:

The manuscript has been significantly improved since the initial submission, with clearer explanations and refined methodology. However, a major conceptual issue remains unresolved: the justification for selecting silver needle thermal conduction therapy (SNTCT) over conventional dry needling (DN). This aspect must be explicitly addressed throughout the manuscript to enhance its scientific rigor and clinical relevance.

Major Issues:

1. Justification for Silver Needle Thermal Conduction Therapy (SNTCT)

The manuscript should clearly define why the authors chose this method over conventional DN. The following aspects require detailed discussion:

Why introduce silver and heat to the needle when standard fine needles are already effective for trigger point inactivation?

What specific physiological or biomechanical advantages does silver provide in this context? If its antimicrobial properties or conductivity are relevant, these should be supported with evidence.

How does thermal conduction enhance the therapeutic effect beyond what is achieved with precise DN?

Is there prior research supporting the addition of these variables in needling therapy? If not, a stronger theoretical basis should be provided.

If silver is only used as a thermal conductor, why not use another material?

2. Mechanism of Action & Physiological Effects

The study should clarify the hypothesized mechanisms through which silver and heat alter the effects of DN:

The standard effectiveness of DN is largely attributed to precise intervention and eliciting localized twitch responses (LTRs) in trigger points. How do heat and silver modify this response?

Could thermal conduction alter needle sensitivity, thereby influencing tactile feedback and the ability to localize trigger points?

If heat increases circulation or pain modulation, was this measured or supported by prior research?

Were adverse effects (e.g., risk of burns, altered tissue response) considered?

3. Study Design Considerations

To isolate the true effect of silver and heat, the study should ideally include:

A comparison between standard DN, silver-coated DN, and silver-thermal DN to assess individual contributions.

A discussion on whether blinding was possible and how placebo effects were minimized.

If no direct comparison was made, the authors should acknowledge this as a limitation and suggest future studies to address it.

4. Clinical Applicability & Cost-Benefit Considerations

If precise DN is already effective, does the added complexity and potential cost of SNTCT justify its use?

Are there specific clinical scenarios where silver thermal conduction needling would be superior to traditional DN?

Could this method introduce risks or complications that are not present with standard DN?

5. Needling Precision & Local Twitch Responses (LTRs)

The therapeutic effectiveness of DN is highly dependent on precise needling technique and the elicitation of LTRs. The manuscript does not discuss whether silver-thermal needling affects this precision.

The authors should provide detailed descriptions of their needling protocol to ensure consistency with established DN methodologies, such as those described in [https://pmc.ncbi.nlm.nih.gov/articles/PMC3533862/].

If LTRs were monitored, this should be explicitly stated. If not, the authors should acknowledge that this is a limitation of the study.

Minor Issues:

Terminology consistency: Ensure that all references to the intervention (SNTCT, silver-thermal DN, etc.) are consistent throughout the manuscript.

Literature review expansion: Include more references to existing studies comparing different needling techniques.

Clarify study limitations: If the study does not separate silver and heat effects, this should be acknowledged.

Conclusion:

The manuscript must provide a strong, evidence-based justification for incorporating silver and heat into DN. Additionally, the study design should either control for these variables separately or acknowledge this as a limitation. Addressing these concerns will significantly enhance the manuscript’s scientific and clinical relevance.

---

## [Author Response · Author response to Decision Letter 3]

18 Jun 2025

Thank you very much for the valuable comments, which have been very helpful for our research and manuscript. We have read the reviewers' opinions and we find that the reviewers' revision suggestions this time are the same as last time. We have once again explained the reviewers' questions in this revised draft. If you have any questions, please contact us at any time. We have responded and made modifications to your question, as follows.

Major Issues:

1. Justification for Silver Needle Thermal Conduction Therapy (SNTCT)

The manuscript should clearly define why the authors chose this method over conventional DN. The following aspects require detailed discussion:

Why introduce silver and heat to the needle when standard fine needles are already effective for trigger point inactivation?

What specific physiological or biomechanical advantages does silver provide in this context? If its antimicrobial properties or conductivity are relevant, these should be supported with evidence.

How does thermal conduction enhance the therapeutic effect beyond what is achieved with precise DN?

Is there prior research supporting the addition of these variables in needling therapy? If not, a stronger theoretical basis should be provided.

If silver is only used as a thermal conductor, why not use another material?

Reply Thank you very much for your valuable comments. First of all, silver needles and dry needles are very different: 1, the treatment site is different: silver needles are based on the anatomical structure of the pain site, and the treatment is performed at the starting point, stopping point or shape of the muscle, while dry needles are treated according to the acupuncture point; 2. Different acupuncture depth: the depth of the silver needle reaches the periosteal, and the action site of the dry needle is relatively shallow; 3. Different thickness: the silver needle body is thicker, the diameter is about 1.1mm, the length is between 110-150mm, the dry needle diameter is about 0.3-0.5mm, the length is about 40-50mm [3, 4], the length and diameter of the silver needle is about 3 times that of the dry needle; 4. Different treatment times: silver needle thermal conduction therapy only requires a single treatment, while dry needle therapy usually requires regular multiple treatments, as seen in page 3, line 46.

Secondly, research indicates that dry acupuncture has poor long-term results [16, 17]. For patients experiencing extreme pain, the impact is greatly diminished[18]. The clinical advantages of dry acupuncture in improving functional disorders and its follow-up effects remain unclear [19].When it comes to improving motor function and everyday life activities and reducing spasticity,warm acupuncture is better than electroacupuncture or acupuncture[27]. The clinical effect of fire acupuncture in treating spasticity after stroke is superior to that of traditional acupunctur[28]. The therapeutic effect of intensive moxibustion plus acupuncture is superior to that of simple acupuncture in improving symptoms of frozen shoulder in patients [29]. It is suggested that thermal effect combined with mechanical effect is superior to mechanical effect alone. silver needle thermal conduction therapy has both thermal and mechanical effects, as seen in page 14, line 323 and page 14, line 341.

Silver has superior thermal conductivity compared to other metals, which is why silver needles are utilized[30]. However, because the application of silver needle thermal conduction therapy is equipped with heating equipment that has the function of temperature control, and we control the temperature of the skin injection point of the silver needle below 42℃, there is no need to worry about side effects like scalding caused by heating. The safety of this treatment has been validated by numerous Chinese literatures, as seen in page 14, line 346.

References:

[3]Ziaeifar M, Arab AM, Mosallanezhad Z, Nourbakhsh MR. Dry needling versus trigger point compression of the upper trapezius: a randomized clinical trial with two-week and three-month follow-up. J Man Manip Ther. 2019 Jul;27(3):152-161.

[4]Jo HR, Choi SK, Sung WS, Lee SD, Lee BW, Kim EJ. Thermal Properties of Warm- versus Heated-Needle Acupuncture. Evid Based Complement Alternat Med. 2022 Feb 28;2022:4159172.

[16]Rodríguez-Mansilla J, González-Sánchez B, De Toro García Á, Valera-Donoso E, Garrido-Ardila EM, Jiménez-Palomares M, González López-Arza MV. Effectiveness of dry needling on reducing pain intensity in patients with myofascial pain syndrome: a Meta-analysis. J Tradit Chin Med. 2016 Feb;36(1):1-13.

[17] Gattie E, Cleland JA, Snodgrass S. The Effectiveness of Trigger Point Dry Needling for Musculoskeletal Conditions by Physical Therapists: A Systematic Review and Meta-analysis. J Orthop Sports Phys Ther. 2017 Mar;47(3):133-149. [18]Gerber LH, Sikdar S, Aredo JV, Armstrong K, Rosenberger WF, Shao H, Shah JP. Beneficial Effects of Dry Needling for Treatment of Chronic Myofascial Pain Persist for 6 Weeks After Treatment Completion. PM R. 2017 Feb;9(2):105-112.

[19]Liu L, Huang QM, Liu QG, Thitham N, Li LH, Ma YT, Zhao JM. Evidence for Dry Needling in the Management of Myofascial Trigger Points Associated With Low Back Pain: A Systematic Review and Meta-Analysis. Arch Phys Med Rehabil. 2018 Jan;99(1):144-152.e2.

[27]Yang L, Tan JY, Ma H, Zhao H, Lai J, Chen JX, Suen LKP. Warm-needle moxibustion for spasticity after stroke: A systematic review of randomized controlled trials. Int J Nurs Stud. 2018 Jun;82:129-138.

[28]Qiu X, Gao Y, Zhang Z, Cheng S, Zhang S. Fire Acupuncture versus conventional acupuncture to treat spasticity after stroke: A systematic review and meta-analysis. PLoS One. 2021 Apr 9;16(4):e0249313.

[29]Gao L, Li X, Wang DB, Du ML, Xie J, Gao XY. [Clinical trial of treatment of frozen shoulder by intensive moxibustion plus acupuncture]. Zhen Ci Yan Jiu. 2019 Apr 25;44(4):297-301. Chinese.

[30]Lin L, Cheng K, Shen XY. [Analysis on the temperature-time curve in warm needling manipulation with acupuncture needles of different materials]. Zhongguo Zhen Jiu. 2019 Dec 12;39(12):1301-7. Chinese.

2.Mechanism of Action & Physiological Effects

The study should clarify the hypothesized mechanisms through which silver and heat alter the effects of DN:

The standard effectiveness of DN is largely attributed to precise intervention and eliciting localized twitch responses (LTRs) in trigger points. How do heat and silver modify this response?

Could thermal conduction alter needle sensitivity, thereby influencing tactile feedback and the ability to localize trigger points?

If heat increases circulation or pain modulation, was this measured or supported by prior research?

Were adverse effects (e.g., risk of burns, altered tissue response) considered?

Reply:Thank you very much for your suggestion and we add further clarification to this section.Silver needles may exert therapeutic effects through thermal and mechanical effects. It has been reported in the literature that the needling action mechanically provides a localized stretch to shortened muscle segments and contracted cytoskeletal structures. This would restore the muscle segment to its resting length by reducing the degree of overlap between actin and myosin filaments[20-22]. Silver needle conductive therapy has a needling effect, which may exert its therapeutic effect by relaxing the muscles. Also, deep needling with dry needling has been shown to be more effective than shallow needling in treating pain associated with myofascial trigger points [23]. The depth of action of silver needle thermal conduction therapy reaches the periosteum, which may be one of the mechanisms by which it exerts good therapeutic effects. At the same time, silver needle heat-conducting therapy may exert a thermal effect by transporting temperature to the body. The literature suggests that warm acupuncture is superior to electroacupuncture or moxibustion in reducing spasticity and promoting motor function and activities of daily living. When comparing warm acupuncture to electroacupuncture or moxibustion, the pooled results for spasticity effects and motor function were significant[27]. Compared to traditional acupuncture, fire acupuncture was more clinically beneficial in treating post-stroke spasticity[28].Intensive moxibustion plus acupuncture was superior to simple acupuncture in improving patients' frozen shoulder symptoms [29]. All of the above results suggest that the heat effect plays a role in improving spasticity.Combined with the result that silver needle thermal conduction therapy can improve muscle hardness, we speculate that silver needle thermal conduction therapy may have both mechanical and thermal effects. Its specific mechanism of action needs to be further explored�which can be seen in page14, line 330.

As the needling site, needle depth, and treatment modality (instrument heating, needle retention for 15-20 min, no repeated lifting and insertion, etc.) of silver needle are different from that of dry needle, there is no need to trigger a local twitch response and localize the trigger point during the implementation process. Trigger points are caused by localized muscle contracture, and silver needle thermal conduction therapy may improve muscle spasm through thermal effect, thus exerting analgesic effect instead of simply destroying trigger points directly, as seen in page 13, line 318.

Heat can increase circulation or pain modulation. Previous literature has reported that heat therapy promotes muscle relaxation, enhances blood circulation, and modulates injury receptors [24].An increased muscle temperature can initially cause substantial improvements in force production, faster rates of force generation, relaxation, shortening, and production of power output [25].Moxibustion heating dilates blood vessels, increases blood flow, and improves microcirculation [26]�as seen in page 14, line 336.

The implementation of silver needle thermal therapy is equipped with a heating device that has the function of temperature regulation. We control the temperature of the skin entry point of the silver needle to below 42℃, which does not cause burns or necrosis of the muscle tissue.

Reference:

[20]Dommerholt J. Dry needling in orthopedic physical therapy practice. Orthop Phys Ther Pract. 2004;16 (3):15–20.

[21]Luke DR. Therapeutic needling in osteopathic practice: an evidence-informed perspective. Int J Osteopath Med. 2009;12(1):2–13.

[22]Dommerholt J, Del Moral OM, Gröbli C. Trigger point dry needling. In: Myofascial Trigger Points: pathophysiology and evidence-informed diagnosis and management. Jones & Bartlett Learning, 2009. p. 159–190.

[23]Kalichman L, Vulfsons S. Dry needling in the management of musculoskeletal pain. J Am Board Fam Med. 2010 Sep-Oct;23(5):640-6.

[24]Rossi R. Heat therapy for different knee diseases: expert opinion. Front Rehabil Sci. 2024 Jul 4;5:1390416. doi: 10.3389/fresc.2024.1390416. PMID: 39055174; PMCID: PMC11270809.

3.Study Design Considerations

To isolate the true effect of silver and heat, the study should ideally include:

A comparison between standard DN, silver-coated DN, and silver-thermal DN to assess individual contributions.

A discussion on whether blinding was possible and how placebo effects were minimized.

If no direct comparison was made, the authors should acknowledge this as a limitation and suggest future studies to address it.

Reply: Thank you very much for your valuable comments, which are very important for a deeper comprehension of the mechanisms of silver needle thermal therapy and may even reducing healthcare expenses. The study's shortcomings have been noted and incorporated in the discussion section (as seen in page15, line 365). In the subsequent experiments, we will add groups as per your suggestions to further clarify the thermal effect and needling effect of the silver needle.

4. Clinical Applicability & Cost-Benefit Considerations

If precise DN is already effective, does the added complexity and potential cost of SNTCT justify its use?

Are there specific clinical scenarios where silver thermal conduction needling would be superior to traditional DN?

Could this method introduce risks or complications that are not present with standard DN?

Reply: Long-term effect of dry needling is not good. In our clinical work we have found that silver needle thermal conduction therapy has definite and long term efficacy for MPS patients who are not well treated with dry needling. I myself am a patient of MPS in neck and shoulder, and the pain was relieved briefly after massage, oral medication and dry needling. By chance, I tried silver needle thermal conduction therapy, and after only one treatment, the pain was significantly relieved, and it has been nearly 10 years since then. As a result, I became interested in this folk therapy and conducted preliminary research. At the same time, silver needles can be sterilized and reused repeatedly, and although the input cost is high, it can be used for a long time.

Based on the good long-term efficacy of silver needles and the number of treatments in a single session, I believe that silver needle thermal conduction therapy is especially necessary.

The implementation of silver needle thermal conduction therapy is equipped with a heating device that has the function of temperature regulation, and we control the temperature at the point of entry into the skin of the silver needle to less than 42℃. The safety of this therapy is confirmed by a large number of literatures in our country.

5. Needling Precision & Local Twitch Responses (LTRs)

The therapeutic effectiveness of DN is highly dependent on precise needling technique and the elicitation of LTRs. The manuscript does not discuss whether silver-thermal needling affects this precision.

The authors should provide detailed descriptions of their needling protocol to ensure consistency with established DN methodologies, such as those described in [https://pmc.ncbi.nlm.nih.gov/articles/PMC3533862/].

If LTRs were monitored, this should be explicitly stated. If not, the authors should acknowledge that this is a limitation of the study.

Reply: In fact, the silver needle thermal conduction therapy does not locate based on trigger points, but rather arranges the needles according to the anatomy of the painful area and the tense zone, such as the starting point, ending point or course of the muscle in the painful area. A localized twitch response may be induced during needling. The needling and thermal effects may exert analgesic effect through other pathways than simply destroying the trigger point directly. The fact that it is not easy to precisely localize the trigger point, and it is difficult to implement even under ultrasound guidance, may be one of the reasons why the efficacy of dry needling is unstable and why ultrasound-guided dry needling has not been widely promoted. We describe this in more detail on page 14, line 326. Long-term efficacy monitoring was not performed, which is a shortcoming of this experiment and is described on page 15, line 367.

Minor issue:

Terminology consistency: Ensure that all references to the intervention (SNTCT, silver-thermal DN, etc.) are consistent throughout the manuscript.

Reply We have harmonized this issue.

Literature review expansion: Include more references to existing studies comparing different needling techniques.

Reply The differences between dry needle and silver needle have been explained in more detail in the manuscript�as seen in page13, line 318.

Clarify study limitations: If the study does not separate silver and heat effects, this should be acknowledged.

Reply We recognize this limitation and account for it in the manuscript, as seen in page 15, line 367.

Conclusion:

The manuscript must provide a strong, evidence-based justification for incorporating silver and heat into DN. Additionally, the study design should either control for these variables separately or acknowledge this as a limitation. Addressing these concerns will significantly enhance the manuscript’s scientific and clinical relevance.

Reply Your comments have been very helpful. In the manuscript we did not validate the thermal and pinprick effects of silver needle conductive

---

## [Decision Letter · Decision Letter 3]

2 Jul 2025

PONE-D-24-35346R3

Quantitative evaluation of therapeutic effect of silver needle thermal conduction therapy on myofascial trigger pointPLOS ONE

Dear Dr. Wang,

Thank you for submitting your manuscript to PLOS ONE. After careful consideration, we feel that it has merit but does not fully meet PLOS ONE’s publication criteria as it currently stands. Therefore, we invite you to submit a revised version of the manuscript that addresses the points raised during the review process.

We look forward to receiving your revised manuscript.

Kind regards,

Jiajia Ye

Guest Editor

PLOS ONE

Comments from PLOS Editorial Office:

We note that one or more reviewers has recommended that you cite specific previously published works. As always, we recommend that you please review and evaluate the requested works to determine whether they are relevant and should be cited. It is not a requirement to cite these works. We appreciate your attention to this request.

Journal Requirements:

Reviewers' comments:

Reviewer's Responses to Questions

**Comments to the Author**

1. If the authors have adequately addressed your comments raised in a previous round of review and you feel that this manuscript is now acceptable for publication, you may indicate that here to bypass the “Comments to the Author” section, enter your conflict of interest statement in the “Confidential to Editor” section, and submit your "Accept" recommendation.

Reviewer #2: All comments have been addressed

2. Is the manuscript technically sound, and do the data support the conclusions?

Reviewer #2: Yes

3. Has the statistical analysis been performed appropriately and rigorously? 

Reviewer #2: Yes

4. Have the authors made all data underlying the findings in their manuscript fully available?

Reviewer #2: Yes

5. Is the manuscript presented in an intelligible fashion and written in standard English?

Reviewer #2: Yes

6. Review Comments to the Author

Reviewer #2: The revised manuscript significantly improves on the original submission, offering detailed explanations on the mechanism and application of silver needle thermal conduction therapy (SNTCT) for myofascial pain syndrome (MPS). The use of T2 mapping, sound touch elastography (STE), transmission electron microscopy (TEM), and molecular assays provides a valuable multiparametric approach to objectively assess muscle and mitochondrial recovery. Ethical compliance and experimental methodology are well documented.

However, there remain important issues regarding the characterization of comparative therapies, particularly dry needling (DN), and the limitations of the study design.

Major Strengths

Comprehensive multimodal evaluation including MRI, elastography, EMG, histology, and SIRT3 analysis.

Clear explanation of SNTCT protocol and physiological rationale.

Acknowledgment of study limitations in the discussion section.

Ethically approved and statistically reasonable animal study design.

Key Issues and Recommendations

1. Misrepresentation of Dry Needling

The authors' attempt to differentiate SNTCT from dry needling (DN) is not adequately supported and contains misleading claims. The response (Major Issue 1) suggests that DN is a superficial, acupuncture-point-based technique requiring repeated treatment sessions. This description does not reflect current clinical practice or research evidence.

Modern dry needling—particularly ultrasound-guided dry needling (US-DN)—enables precise targeting of myofascial trigger points (TrPs) under direct anatomical visualization. As shown in publications by Bubnov R. (Bubnov, 2013; https://www.liebertpub.com/doi/abs/10.1089/acu.2013.0973
https://linkinghub.elsevier.com/retrieve/pii/S0003496724543175, etc.), US-DN can be highly effective, reproducible, and minimally invasive, often requiring a single treatment without the need for thermal or material enhancements.

Additionally, the assertion that DN lacks long-term efficacy or acts solely through mechanical disruption without benefit is overly simplistic. DN protocols often elicit local twitch responses (LTRs), a known therapeutic mechanism in TrP inactivation. The role of SNTCT should be discussed as an alternative or adjunct, not a superior replacement, unless comparative data are provided.

Recommendation: The manuscript should be revised to accurately represent DN, particularly ultrasound-guided DN, and provide a balanced comparison. Please acknowledge that:

DN can be anatomically precise.

US guidance enhances safety and efficacy.

DN may not universally require repeated sessions.

Any claimed superiority of SNTCT should be tested in direct comparative studies.

2. Study Design Limitations

The lack of comparator groups (e.g., DN, silver-only, heat-only) prevents conclusions about which component(s) of SNTCT are responsible for observed effects. While acknowledged in the discussion, this limitation is fundamental and should be clearly emphasized in the conclusions.

3. Clinical Translation

Given the strong animal data, the authors should discuss the translational roadmap for SNTCT, including safety, patient selection, cost implications, and potential for integration with ultrasound guidance.

4. Terminology

Please ensure consistent use of terminology (e.g., "silver needle thermal conduction therapy" vs. "SNTCT" vs. "silver-thermal DN").

5. Mitochondrial Mechanism

The link between mitochondrial repair and pain relief is plausible but remains speculative without direct measurement of functional mitochondrial output (e.g., ATP, ROS). This should be acknowledged more explicitly.

Final Recommendation

Minor Revision

The manuscript presents valuable and novel findings. However, it requires clarification and correction of statements about comparative therapies—particularly modern DN techniques—to ensure scientific integrity. A revised conclusion section acknowledging these limitations and a more balanced comparison will significantly strengthen the paper’s clinical relevance and accuracy.

7. PLOS authors have the option to publish the peer review history of their article (what does this mean? ). If published, this will include your full peer review and any attached files.

**Do you want your identity to be public for this peer review?** For information about this choice, including consent withdrawal, please see our Privacy Policy .

Reviewer #2: **Yes: ** Rostyslav Bubnov

---

## [Author Response · Author response to Decision Letter 4]

4 Jul 2025

Thank you very much for the valuable comments, which have been very helpful for our research and manuscript. We have responded and made modifications to your question, as follows.

Key Issues and Recommendations

1. Misrepresentation of Dry Needling

The authors' attempt to differentiate SNTCT from dry needling (DN) is not adequately supported and contains misleading claims. The response (Major Issue 1) suggests that DN is a superficial, acupuncture-point-based technique requiring repeated treatment sessions. This description does not reflect current clinical practice or research evidence.

Modern dry needling—particularly ultrasound-guided dry needling (US-DN)—enables precise targeting of myofascial trigger points (TrPs) under direct anatomical visualization. As shown in publications by Bubnov R. (Bubnov, 2013; https://www.liebertpub.com/doi/abs/10.1089/acu.2013.0973
https://linkinghub.elsevier.com/retrieve/pii/S0003496724543175, etc.), US-DN can be highly effective, reproducible, and minimally invasive, often requiring a single treatment without the need for thermal or material enhancements.

Additionally, the assertion that DN lacks long-term efficacy or acts solely through mechanical disruption without benefit is overly simplistic. DN protocols often elicit local twitch responses (LTRs), a known therapeutic mechanism in TrP inactivation. The role of SNTCT should be discussed as an alternative or adjunct, not a superior replacement, unless comparative data are provided.

Recommendation: The manuscript should be revised to accurately represent DN, particularly ultrasound-guided DN, and provide a balanced comparison. Please acknowledge that:

DN can be anatomically precise.

US guidance enhances safety and efficacy.

DN may not universally require repeated sessions.

Any claimed superiority of SNTCT should be tested in direct comparative studies.

Reply: We have made the necessary revisions to the above issues, as seen in line 55, 320, and 327.

2. Study Design Limitations

The lack of comparator groups (e.g., DN, silver-only, heat-only) prevents conclusions about which component(s) of SNTCT are responsible for observed effects. While acknowledged in the discussion, this limitation is fundamental and should be clearly emphasized in the conclusions.

Reply We have already emphasized this limitation as per your suggestion�as seen in page 15, line 359.

3. Clinical Translation

Given the strong animal data, the authors should discuss the translational roadmap for SNTCT, including safety, patient selection, cost implications, and potential for integration with ultrasound guidance.

Reply: Thank you very much for your valuable comments. Your suggestion is of great help to our subsequent research content. We may conduct further exploration in terms of economic costs and treatment guided by ultrasound. In fact, this technology has been applied in clinical practice for many years. Based on its definite long-term efficacy in clinical practice, our research group conducted some related studies on this therapy. In recent years, for the treatment of some specific areas such as the chest and back, ultrasound guidance has been adopted to enhance safety.

4. Terminology

Please ensure consistent use of terminology (e.g., "silver needle thermal conduction therapy" vs. "SNTCT" vs. "silver-thermal DN").

Reply: We once again checked the consistency of the "silver needle thermal conduction therapy" section in the manuscript.

5. Mitochondrial Mechanism

The link between mitochondrial repair and pain relief is plausible but remains speculative without direct measurement of functional mitochondrial output (e.g., ATP, ROS). This should be acknowledged more explicitly.

Reply: Thank you very much for your valuable comments. The limitation have been stated in the manuscript on page15, line 354.

---

## [Editor Report · Decision Letter 4]

10 Jul 2025

PONE-D-24-35346R4Quantitative evaluation of therapeutic effect of silver needle thermal conduction therapy on myofascial trigger pointPLOS ONE

Dear Dr. Wang,

Thank you for submitting your manuscript to PLOS ONE. After careful consideration, we feel that it has merit but does not fully meet PLOS ONE’s publication criteria as it currently stands. Therefore, we invite you to submit a revised version of the manuscript that addresses the points raised during the review process.

We look forward to receiving your revised manuscript.

Kind regards,

Jiajia Ye

Guest Editor

PLOS ONE

Journal Requirements:

Additional Editor Comments :

It is strongly recommended that the authors carefully revise the page and line numbers to ensure that the editor and reviewers can clearly identify the changes made.

Additionally, please note that the suggested references are optional. Authors are encouraged to thoroughly review and assess the relevance of each suggestion, and only cite references that are highly pertinent to the manuscript.

---

## [Author Response · Author response to Decision Letter 5]

16 Jul 2025

Thank you very much for your comments and suggestions on the manuscript, which have helped us present the content of the manuscript more precisely.While, we sincerely apologize for the inconvenience caused to you due to unclear identification of the modified content.We have responded and made modifications to your question, as follows.

Key Issues and Recommendations

1. Misrepresentation of Dry Needling

The authors' attempt to differentiate SNTCT from dry needling (DN) is not adequately supported and contains misleading claims. The response (Major Issue 1) suggests that DN is a superficial, acupuncture-point-based technique requiring repeated treatment sessions. This description does not reflect current clinical practice or research evidence.

Modern dry needling—particularly ultrasound-guided dry needling (US-DN)—enables precise targeting of myofascial trigger points (TrPs) under direct anatomical visualization. As shown in publications by Bubnov R. (Bubnov, 2013; https://www.liebertpub.com/doi/abs/10.1089/acu.2013.0973
https://linkinghub.elsevier.com/retrieve/pii/S0003496724543175, etc.), US-DN can be highly effective, reproducible, and minimally invasive, often requiring a single treatment without the need for thermal or material enhancements.

Additionally, the assertion that DN lacks long-term efficacy or acts solely through mechanical disruption without benefit is overly simplistic. DN protocols often elicit local twitch responses (LTRs), a known therapeutic mechanism in TrP inactivation. The role of SNTCT should be discussed as an alternative or adjunct, not a superior replacement, unless comparative data are provided.

Recommendation: The manuscript should be revised to accurately represent DN, particularly ultrasound-guided DN, and provide a balanced comparison. Please acknowledge that:

DN can be anatomically precise.

US guidance enhances safety and efficacy.

DN may not universally require repeated sessions.

Any claimed superiority of SNTCT should be tested in direct comparative studies.

Reply: We have made the necessary revisions to the above issues, as seen in line 340-345, page 14.

2.Study Design Limitations

The lack of comparator groups (e.g., DN, silver-only, heat-only) prevents conclusions about which component(s) of SNTCT are responsible for observed effects. While acknowledged in the discussion, this limitation is fundamental and should be clearly emphasized in the conclusions.

Reply We have already emphasized this limitation as per your suggestion�as seen in page 15, line 364-367.

3. Clinical Translation

Given the strong animal data, the authors should discuss the translational roadmap for SNTCT, including safety, patient selection, cost implications, and potential for integration with ultrasound guidance.

Reply: Thank you very much for your valuable comments. Your suggestion is of great help to our subsequent research content. We may conduct further exploration in terms of economic costs and treatment guided by ultrasound. In fact, this technology has been applied in clinical practice for many years. Based on its definite long-term efficacy in clinical practice, our research group conducted some related studies on this therapy. In recent years, for the treatment of some specific areas such as the chest and back, ultrasound guidance has been adopted to enhance safety.

4. Terminology

Please ensure consistent use of terminology (e.g., "silver needle thermal conduction therapy" vs. "SNTCT" vs. "silver-thermal DN").

Reply: We once again checked the consistency of the "silver needle thermal conduction therapy" section in the manuscript.

5. Mitochondrial Mechanism

The link between mitochondrial repair and pain relief is plausible but remains speculative without direct measurement of functional mitochondrial output (e.g., ATP, ROS). This should be acknowledged more explicitly.

Reply: Thank you very much for your valuable comments. The limitation have been stated in the manuscript on page15, line 358-360.

---

## [Decision Letter · Decision Letter 5]

5 Aug 2025

Quantitative evaluation of therapeutic effect of silver needle thermal conduction therapy on myofascial trigger point

PONE-D-24-35346R5

Dear Dr. Wang,

We’re pleased to inform you that your manuscript has been judged scientifically suitable for publication and will be formally accepted for publication once it meets all outstanding technical requirements.

Kind regards,

Jiajia Ye

Guest Editor

PLOS ONE

Additional Editor Comments (optional):

Reviewers' comments:

Reviewer's Responses to Questions

**Comments to the Author**

1. If the authors have adequately addressed your comments raised in a previous round of review and you feel that this manuscript is now acceptable for publication, you may indicate that here to bypass the “Comments to the Author” section, enter your conflict of interest statement in the “Confidential to Editor” section, and submit your "Accept" recommendation.

Reviewer #2: All comments have been addressed

2. Is the manuscript technically sound, and do the data support the conclusions?

Reviewer #2: Yes

3. Has the statistical analysis been performed appropriately and rigorously? 

Reviewer #2: Yes

4. Have the authors made all data underlying the findings in their manuscript fully available?

Reviewer #2: Yes

5. Is the manuscript presented in an intelligible fashion and written in standard English?

Reviewer #2: Yes

6. Review Comments to the Author

Reviewer #2: This version meets the journal’s publication standards. It offers a well-structured preclinical study with clear scientific rationale, multiparametric objective measurements, and an appropriately framed interpretation of results. The authors have shown responsiveness to peer review and improved the manuscript accordingly.

7. PLOS authors have the option to publish the peer review history of their article (what does this mean? ). If published, this will include your full peer review and any attached files.

**Do you want your identity to be public for this peer review?** For information about this choice, including consent withdrawal, please see our Privacy Policy .

Reviewer #2: **Yes: ** Rostyslav Bubnov

---

## [Editor Report · Acceptance letter]

PONE-D-24-35346R5

PLOS ONE

Dear Dr. Wang,

I'm pleased to inform you that your manuscript has been deemed suitable for publication in PLOS ONE. Congratulations! Your manuscript is now being handed over to our production team.

Kind regards,

on behalf of

Dr. Jiajia Ye

Guest Editor

PLOS ONE